# Decoding subjective emotional arousal from EEG during an immersive virtual reality experience

**Simon M Hofmann**[1]*[†], **Felix Klotzsche**[1,2]*[†], **Alberto Mariola**[3,4]*[†], **Vadim Nikulin**[1,5], **Arno Villringer**[1,2], **Michael Gaebler**[1,2]*

[1]Department of Neurology, Max Planck Institute for Human Cognitive and Brain Sciences, Leipzig, Germany; [2]Humboldt-Universität zu Berlin, Faculty of Philosophy, Berlin School of Mind and Brain, Berlin, Germany; [3]Sackler Centre for Consciousness Science, School of Engineering and Informatics, University of Sussex, Brighton, United Kingdom; [4]Sussex Neuroscience, School of Life Sciences, University of Sussex, Brighton, United Kingdom; [5]Bernstein Center for Computational Neuroscience Berlin, Berlin, Germany

**\*For correspondence:**
simon.hofmann@cbs.mpg.de (SMH);
klotzsche@cbs.mpg.de (FK);
a.mariola@sussex.ac.uk (AM);
gaebler@cbs.mpg.de (MG)

[†]These authors contributed equally to this work

**Competing interest:** The authors declare that no competing interests exist.

**Abstract** Immersive virtual reality (VR) enables naturalistic neuroscientific studies while maintaining experimental control, but dynamic and interactive stimuli pose methodological challenges. We here probed the link between emotional arousal, a fundamental property of affective experience, and parieto-occipital alpha power under naturalistic stimulation: 37 young healthy adults completed an immersive VR experience, which included rollercoaster rides, while their EEG was recorded. They then continuously rated their subjective emotional arousal while viewing a replay of their experience. The association between emotional arousal and parieto-occipital alpha power was tested and confirmed by (1) decomposing the continuous EEG signal while maximizing the comodulation between alpha power and arousal ratings and by (2) decoding periods of high and low arousal with discriminative common spatial patterns and a long short-term memory recurrent neural network. We successfully combine EEG and a naturalistic immersive VR experience to extend previous findings on the neurophysiology of emotional arousal towards real-world neuroscience.

## Editor's evaluation

Hofmann et al., investigate the link between two phenomena, emotional arousal and cortical α activity. Although α activity is tightly linked to the first reports of electric activity in the brain nearly 100 years ago, a comprehensive characterization of this phenomenon is elusive. One of the reasons is that EEG, the major method to investigate electric activity in the human brain, is susceptible to motion artifacts and, thus, mostly used in laboratory settings. Here, the authors combine EEG with virtual reality (VR) to give experimental participants a roller coaster ride with high immersion. The ride, literally, leads to large ups and downs in emotional arousal, which is quantified by the subjects during a later rerun. Several different decoding methods were evaluated, and each showed above-chance levels of performance, substantiating a link between lower levels of parietal/occipital α and subjective arousal in a quasi-naturalistic setting.

## Introduction

While humans almost constantly interact with complex, dynamic environments, lab-based studies typically use simplified stimuli in passive experimental situations. Trading realism for experimental control

**eLife digest** Human emotions are complex and difficult to study. It is particularly difficult to study emotional arousal, this is, how engaging, motivating, or intense an emotional experience is. To learn how the human brain processes emotions, researchers usually show emotional images to participants in the laboratory while recording their brain activity. But viewing sequences of photos is not quite like experiencing the dynamic and interactive emotions people face in everyday life.

New technologies, such as immersive virtual reality, allow individuals to experience dynamic and interactive situations, giving scientists the opportunity to study human emotions in more realistic settings. These tools could lead to new insights regarding emotions and emotional arousal.

Hofmann, Klotzsche, Mariola et al. show that virtual reality can be a useful tool for studying emotions and emotional arousal. In the experiment, 37 healthy young adults put on virtual reality glasses and 'experienced' two virtual rollercoaster rides. During the virtual rides, Hofmann, Klotzsche, Mariola et al. measured the participants' brain activity using a technique called electroencephalography (EEG). Then, the participants rewatched their rides and rated how emotionally arousing each moment was. Three different computer modeling techniques were then used to predict the participant's emotional arousal based on their brain activity.

The experiments confirmed the results of traditional laboratory experiments and showed that the brain's alpha waves can be used to predict emotional arousal. This suggests that immersive virtual reality is a useful tool for studying human emotions in circumstances that are more like everyday life. This may make future discoveries about human emotions more useful for real-life applications such as mental health care.

happens at the expense of the representativity of the experimental design (*Brunswik, 1955*), that is, the degree to which effects found in the lab generalize to practical everyday-life conditions. This may be particularly true for affective phenomena like emotions.

## Emotional arousal as a fundamental property of affective experience

Emotions are subjective, physiological, and behavioural responses to personally meaningful external stimuli (*Mauss and Robinson, 2009*) or self-generated mental states (e.g., memories; *Damasio et al., 2000*) and underlie our experience of the world (*James, 1884*; *James, 1890*; *Seth, 2013*). Emotions are crucial for physical and mental health (*Gross and Muñoz, 1995*) and their investigation has long been at the core of experimental psychology (*Wundt and Judd, 1897*).

Dimensional accounts conceptualize emotions along the two axes of valence and arousal (*Duffy, 1957*; *Kuppens et al., 2013*; *Russell, 1980*; *Russell and Barrett, 1999*; *Wundt and Judd, 1897*): valence differentiates states of pleasure and displeasure, while emotional arousal describes the degree of activation or intensity that accompanies an emotional state. [Different types of arousal have been proposed and investigated, such as sexual, autonomic, emotional (*Russell, 1980*); also in the context of altered states of consciousness, for example, through anaesthesia or sleep. They may share psychological (e.g., increase in sensorimotor and emotional reactivity; *Pfaff et al., 2012*) and physiological aspects (e.g., sympathetic activation) but are not synonymous. We here explicitly refer to arousal in the context of the subjective experience of emotions].

Emotions have been linked to activity in the autonomic (ANS) and the central nervous system (*Dalgleish, 2004*). It has thereby been difficult to consistently associate individual, discrete emotion categories with specific response patterns in the ANS (*Kragel and Labar, 2013*; *Kreibig, 2010*; *Siegel et al., 2018*) or in distinct brain regions (*Lindquist et al., 2012*; but *Saarimäki et al., 2016*). Rather, emotions seem to be dynamically implemented by sets of brain regions and bodily activations that are involved in basic, also non-emotional psychological operations (i.e., 'psychological primitives'; *Lindquist et al., 2012*). In this view, humans are typically in fluctuating states of pleasant or unpleasant arousal ('core affect'; *Russell and Barrett, 1999*; *Lindquist, 2013*), which can be influenced by external stimuli. Emotional arousal could thereby be a 'common currency' to compare different stimuli or events (*Lindquist, 2013*) and represent fundamental neural processes that underlie a variety of emotions (*Wilson-Mendenhall et al., 2013*). It can fluctuate quickly – on the order of minutes (*Kuppens et al., 2010*) or seconds (*Mikutta et al., 2012*) – and has been connected to

ANS activity, as measured by pupil diameter (*Bradley et al., 2008*) or skin conductance (*Bach et al., 2010*). At the brain level, emotional arousal was linked to lower alpha power, particularly over parietal electrodes (*Luft and Bhattacharya, 2015*; *Koelstra et al., 2012*). The parieto-occipital alpha rhythm, typically oscillating in the frequency range of 8–13 Hz, is the dominant EEG rhythm in awake adults with eyes closed (*Berger, 1929*), where it varies with vigilance (*Olbrich et al., 2009*). However, in tasks of visual processing (i.e., with eyes open), parieto-occipital alpha power was linked to active attentional processes (e.g., distractor suppression; *Kelly et al., 2006*; *Klimesch, 2012*) or, more generally, to functional inhibition for information gating (*Jensen and Mazaheri, 2010*). Physiologically, alpha oscillations were associated with large-scale synchronization of neuronal activity (*Buzsáki, 2006*) and metabolic deactivation (*Moosmann et al., 2003*).

In sum, bodily responses interact in complex ways across situations, and activity in the brain is central for emotions and their subjective component (*Barrett, 2017*; *Seth, 2013*). As arousal is a fundamental property not only of emotions but of subjective experience in general (*Adolphs et al., 2019*), an investigation of its neurophysiology, reflected in neural oscillations, is essential to understanding the biology of the mind.

## Studying emotional arousal and its neurophysiology in the lab

Studies that investigated emotions or emotional arousal in laboratory environments typically used static images. For example, more emotionally arousing relative to less emotionally arousing (e.g., neutral) pictures were associated with an event-related desynchronization, that is, a decrease in the power of alpha oscillations in posterior channels (*De Cesarei and Codispoti, 2011*; *Schubring and Schupp, 2019*; but *Uusberg et al., 2013*). In a study, in which emotional arousal was induced through pictures and music, blocks of higher emotional arousal were associated with decreased alpha power compared to blocks of lower emotional arousal (*Luft and Bhattacharya, 2015*). However, emotion-eliciting content that is repeatedly presented in trials creates an artificial experience for participants (*Bridwell et al., 2018*); it hardly resembles natural human behaviour and its (neuro-)physiology, which unfolds over multiple continuous timescales (*Huk et al., 2018*). Moreover, such presentations lack a sense of emotional continuity. External events often do not appear suddenly but are embedded in an enduring sequence, in which emotions build up and dissipate. Real-life scenarios also include anticipatory aspects where emotional components can be amplified or even suppressed, thus rendering the relationship between the corresponding neuronal events and subjective experience more complex than the one typically studied with randomized or partitioned presentations of visual or auditory stimuli.

Virtual reality (VR) technology – particularly immersive VR, in which the user is completely surrounded by the virtual environment – affords the creation and presentation of computer-generated scenarios that are contextually rich and engaging (*Diemer et al., 2015*). As more naturalistic (i.e., dynamic, interactive, and less decontextualized) experiments allow to study the brain under conditions it was optimized for (*Gibson, 1978*; *Hasson et al., 2020*), their findings may more readily generalize to real-world circumstances and provide better models of the brain (*Matusz et al., 2019*; *Shamay-Tsoory and Mendelsohn, 2019*).

In this study, we aimed to link subjective emotional arousal with alpha power in a naturalistic setting. Participants completed an immersive VR experience that included virtual rollercoaster rides while their EEG was recorded. They then continuously rated their emotional arousal while viewing a replay of their previous experience (*McCall et al., 2015*).

## Methodological challenges of naturalistic experiments

To tackle the challenges of data acquired in naturalistic settings and with continuous stimuli, we made use of recent advances in signal processing and statistical modelling: spatial filtering methods (originally developed for brain-computer interfaces [BCIs]; *Blankertz et al., 2008*) have recently gained popularity in cognitive neuroscience (*Cohen, 2018*; *Zuure and Cohen, 2020*), where they have been used to analyze continuous data collected in naturalistic experiments, for example, to find inter-subject correlations in neuroimaging data of participants watching the same movie (*Dmochowski et al., 2012*; *Gaebler et al., 2014*).

For the present experiment, two spatial filtering methods were applied to link alpha power and subjective emotional arousal: source power comodulation (SPoC; *Dähne et al., 2014*) and common spatial patterns (CSP; *Blankertz et al., 2008*; *Ramoser et al., 2000*).

SPoC is a supervised regression approach, in which a target variable (here: subjective emotional arousal) guides the extraction of relevant M/EEG oscillatory components (here: alpha power). SPoC has been used to predict single-trial reaction times from alpha power in a hand motor task (*Meinel et al., 2016*), muscular contraction from beta power (*Sabbagh et al., 2020*), and difficulty levels of a video game from theta and alpha power (*Naumann et al., 2016*). CSP is used to decompose a multivariate signal into components that maximize the difference in variance between distinct classes (here: periods of high and low emotional arousal). CSP thereby allows optimizing the extraction of power-based features from oscillatory signals, which can then be applied for training classifiers to solve binary or categorical prediction problems. CSP is being used with EEG for BCI (*Blankertz et al., 2008*) or to decode workload (*Schultze-Kraft et al., 2016*).

In addition to M/EEG-specific spatial filtering methods, non-linear machine learning methods are suited for the analysis of continuous, multidimensional recordings from naturalistic experiments. Deep neural networks transform high-dimensional data into target output variables (here: different states of emotional arousal) by finding statistical invariances and hidden representations in the input (*Goodfellow et al., 2016*; *LeCun et al., 2015*; *Schmidhuber, 2015*). For time-sequential data, long short-term memory (LSTM) recurrent neural networks (RNNs) are particularly suited (*Greff et al., 2017*; *Hochreiter and Schmidhuber, 1995*; *Hochreiter and Schmidhuber, 1997*). Via nonlinear gating units, the LSTM determines which information flows in and out of the memory cell in order to find long- and short-term dependencies over time. LSTMs have been successfully applied for speech recognition (*Graves et al., 2013*), language translation (*Luong et al., 2015*), or scene analysis in videos (*Donahue et al., 2015*), but also to detect emotions in speech and facial expressions (*Wöllmer et al., 2010*; *Wöllmer et al., 2008*) or workload in EEG (*Bashivan et al., 2016*; *Hefron et al., 2017*). In comparison to other deep learning methods, LSTMs are 'quick learners' due to their efficient gradient flow and thus suitable for the continuous and sparse data recorded under naturalistic stimulation with VR.

The present study tested the hypothesis of a negative association between parieto-occipital alpha power and subjective emotional arousal under dynamic and interactive stimulation. Combining immersive VR and EEG, this study aimed to (1) induce variance in emotional arousal in a naturalistic setting and (2) capture the temporally evolving and subjective nature of emotional arousal via continuous ratings in order to (3) assess their link to oscillations of brain activity in the alpha frequency range. The link between subjective emotional arousal and alpha power was then tested by decoding the former from the latter using the three complementary analysis techniques SPoC, CSP, and LSTM.

# Materials and methods

## Participants

Forty-five healthy young participants were recruited via the participant database at the Berlin School of Mind and Brain (an adaption of ORSEE; *Greiner, 2015*). Previous studies on the relationship between emotional arousal and neural oscillations reported samples of 19–32 subjects (e.g., *Koelstra et al., 2012*; *Luft and Bhattacharya, 2015*). We recruited more participants to compensate for anticipated dropouts due to the VR setup and to ensure a robust estimate of the model performances. Inclusion criteria were right-handedness, normal or corrected-to-normal vision, proficiency in German, no (self-reported) psychiatric, or neurological diagnoses in the past 10 years, and less than 3 hr of experience with VR. Participants were requested to not drink coffee or other stimulants 1 hr before coming to the lab. The experiment took ~2.5 hr, and participants were reimbursed with 9€ per hour. They signed informed consent before their participation, and the study was approved by the Ethics Committee of the Department of Psychology at the Humboldt-Universität zu Berlin.

## Setup, stimuli, and measures

The experiment was conducted in a quiet room, in which the temperature was kept constant at 24 °C.

### Neurophysiology/EEG

Thirty channels of EEG activity were recorded in accordance with the international 10/20 system (*Sharbrough et al., 1991*) using a mobile amplifier (LiveAmp32) and active electrodes (actiCap; both by BrainProducts, Gilching, Germany, RRID:SCR_009443). Two additional electrooculogram (EOG) electrodes were placed below and next to the right eye to track eye movements. Data were sampled at

500 Hz with a hardware-based low-pass filter at 131 Hz and referenced to electrode FCz. The amplifier was placed on a high table in the back of the participant to minimize the pull on electrode cables and provide maximal freedom for head movements. The VR headset was placed carefully on top of the EEG cap, and impedances were brought below 10 kΩ. With the same amplifier, electrocardiography and galvanic skin responses were additionally acquired. These peripheral physiological data and the inter-individual differences in interoceptive accuracy are beyond the scope of this paper, and their results will be reported elsewhere.

### VR HMD
An HTC Vive head-mounted display (HMD; HTC, New Taipei, Taiwan) and headphones (AIAIAI Tracks, ApS, Copenhagen, Denmark) were placed on top of the EEG cap using small, custom-made cushions to avoid pressure artefacts and increase comfort. The HTC Vive provides stereoscopy with two 1080 × 1200-pixel OLED displays, a 110° field-of-view, and a frame rate of 90 Hz. The user's head position is tracked using infrared light, accelerometry, and gyroscopy. Head movements were recorded by adapting scripts from *Thor, 2016*.

### Immersive VR experience/stimulation
Stimulation comprised two commercially available rollercoaster rides ('Russian VR Coasters' by Funny Twins Games, Ekaterinburg, Russia, on Steam) that were separated by a 30 s break (during which participants kept their eyes open and looked straight): the 'Space' rollercoaster, a 153 s ride through planets, asteroids, and spaceships and the 'Andes' rollercoaster, a 97 s ride through a mountain scenery (for more details, see Figure 5 and the Appendix 1). The two rollercoaster rides were commercially available on Steam. The rollercoasters were selected for their length (to not cause physical discomfort by wearing the HMD for too long) and content (to induce variance in emotional arousal). The experience, comprising the sequence 'Space'-break-'Andes', was kept constant across participants.

## Self-reports
### Questionnaires
At the beginning of the experiment, participants completed two arousal-related questionnaires: (1) the 'Trait' subscale of the 'State-Trait Anxiety Inventory' (STAI-T; *Spielberger, 1983*; *Spielberger, 1989*) and (2) the 'Sensation Seeking' subscale of the 'UPPS Impulsive Behaviour Scale' (UPPS; *Schmidt et al., 2008*; *Whiteside and Lynam, 2001*). Before and after the experiment, participants completed a customized version of the 'Simulator Sickness Questionnaire' (SSQ, *Bouchard et al., 2017*) comprising three items from the nausea (general discomfort, nausea, dizziness) and three items

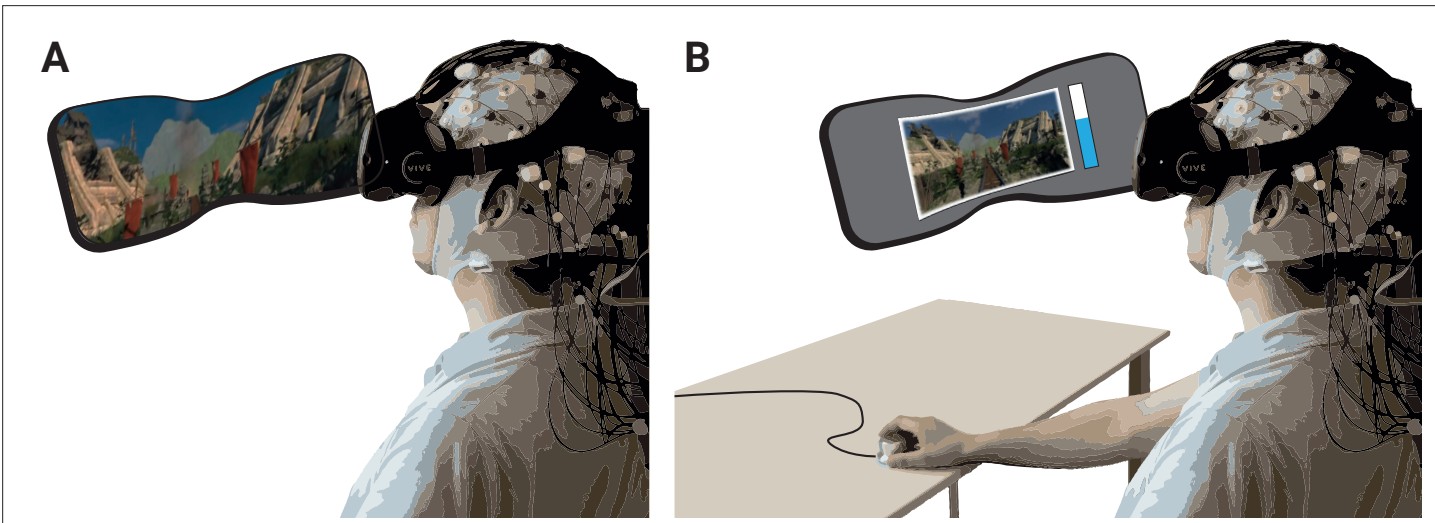

**Figure 1.** Schematic of experimental setup. (**A**) The participants underwent the experience (two rollercoasters separated by a break) in immersive virtual reality (VR), while EEG was recorded. (**B**) They then continuously rated the level of emotional arousal with a dial viewing a replay of their experience. The procedure was completed twice, without and with head movements.

from the oculomotor subscale (headache, blurred vision, difficulty concentrating) to capture potential VR side effects (*Sharples et al., 2008*). After the experiment, participants also rated the presence and valence of their experience (the results will be reported elsewhere).

### Emotional arousal

After each VR experience, participants watched a 2D recording (recorded using OBS Studio, https://obsproject.com/) of their experience on a virtual screen (SteamVR's 'view desktop' feature), that is, without removing the HMD. They recalled and continuously rated their emotional arousal by turning a dial (PowerMate USB, Griffin Technology, Corona, CA; sampling frequency: 50 Hz), with which they manipulated a vertical rating bar, displayed next to the video, ranging from low (0) to high (100) in 50 discrete steps (*McCall et al., 2015*; see *Figure 1B*). The exact formulation was 'When we show you the video, please state continuously how emotionally arousing or exciting the particular moment during the VR experience was' (German: 'Wenn wir dir das Video zeigen, gebe bitte durchgehend an, wie emotional erregend, bzw. aufregend der jeweilige Moment während der VR Erfahrung war'). To present the playback video and the rating bar, a custom script written in Processing (v3.0) was used.

### Procedure

Participants came to the lab and filled in the pre-test questionnaires. After the torso and limb electrodes had been attached, participants completed a heartbeat guessing task (*Schandry, 1981*) to assess inter-individual differences in interoceptive accuracy (the results of peripheral physiology and interoception will be reported elsewhere). Then, the EEG cap was attached, and the HMD was carefully placed on top of it. To prevent or minimize (e.g., movement-related) artefacts, customized cushions were placed below the straps of the VR headset to reduce the contact with the EEG sensors. In addition, the VR experience took place while seated and without full body movements (participants were asked to keep their feet and arms still during the recordings). A white grid was presented in the HMD to ensure that the participants' vision was clear. They then completed a 10 min resting-state phase (5 min eyes open, 5 min eyes closed), before experiencing the first VR episode, which consisted of the two virtual rollercoaster rides and the intermediate break: first the 'Space' and then, after the break, the 'Andes' rollercoaster. In the subsequent rating phase, they recalled and continuously rated their emotional arousal while viewing a 2D recording of their experience. Importantly, each participant completed the VR episode (plus rating) twice: once while not moving the head (*nomov* condition) and once while freely moving the head (*mov* condition) during the VR experience. The sequence of the movement conditions was counterbalanced across participants (*n* = 19 with nomov condition first). At the end of the experiment, participants completed two additional questionnaires (the SUS and the questionnaire on subjective feelings of presence and valence during the virtual rollercoaster rides) before they were debriefed.

### Data analysis

To exclude effects related to the on- or offset of the rollercoasters, data recorded during the first and the last 2.5 s of each rollercoaster were removed and the inter-individually slightly variable break was cropped to 30 s. The immersive VR experience that was analysed thus consisted of two time series of 270 s length each per participant (nomov and mov).

### Self-reports
#### Questionnaires
Inter-individual differences as assessed by the trait questionnaires were not the focus of this study, and their results (together with the peripheral physiological and interoception data) will be reported elsewhere. The sum of the simulator sickness ratings before and after the experiment was compared using a two-sided paired *t*-test.

### Emotional arousal

Emotional arousal ratings were resampled to 1 Hz by averaging non-overlapping sliding windows, yielding one arousal value per second. For the classification analyses, ratings were divided by a tertile split into three distinct classes of arousal ratings (low, medium, high) per participant. For the binary classification (high vs. low arousal), the medium arousal ratings were discarded.

## Neurophysiology

### Preprocessing

EEG data were preprocessed and analyzed with custom MATLAB (RRID:SCR_001622) scripts built on the EEGLAB toolbox (RRID:SCR_007292, v13.5.4b; *Delorme and Makeig, 2004*). The preprocessing steps were applied separately for data recorded during the nomov and mov conditions (i.e., without and with head movement). Continuous data were downsampled to 250 Hz (via the 'pop_resample.m' method in EEGLAB) and PREP pipeline (v0.55.2; *Bigdely-Shamlo et al., 2015*) procedures were applied for detrending (1 Hz high-pass filter, Hamming windowed zero-phase sinc FIR filter, cutoff frequency (–6 dB): 0.5 Hz, filter order: 827, transition band width: 1 Hz), line-noise removal (line frequency: 50 Hz), robust referencing to average, and detection as well as spherical interpolation of noisy channels. Due to the relatively short lengths of the time series, the default fraction of bad correlation windows (parameter 'badTimeThreshold', used to mark bad channels) was increased to 0.05. For all other parameters, default values of PREP were kept. On average, 2.08 and 2.47 channels per subject were interpolated in the nomov and mov condition, respectively. Data remained high-pass filtered for the further steps of the analysis. Retrospective arousal ratings were added to the data sets, labelling each second of data with an associated arousal rating used as target for the later classification and regression approaches.

ICA decomposition was used to identify and remove EEG artefacts caused by eye movements, blinks, and muscular activity. To facilitate the decomposition, ICA projection matrices were calculated on a subset of the data from which the noisiest parts had been removed. To this end, a copy of the continuous data was split into 270 epochs of 1 s length. Epochs containing absolute voltage values > 100 µV in at least one channel (excluding channels that reflected eye movements, i.e., EOG channels, Fp1, Fp2, F7, F8) were deleted. Extended infomax (*Lee et al., 1999*) ICA decomposition was calculated on the remaining parts of the data (after correcting for rank deficiency with a principal component analysis). Subjects with >90 to-be-deleted epochs (33 % of the data) were discarded from further analyses (nomov: $n = 5$; mov: $n = 10$). Artefactual ICA components were semi-automatically selected using the SASICA extension (*Chaumon et al., 2015*) of EEGLAB and visual inspection. On average, 13.41 (nomov) and 10.31 (mov) components per subject were discarded. The remaining ICA weights were back-projected onto the continuous time series.

## Dimensionality reduction: SSD in the (individual) alpha frequency range

Our main hypothesis was that EEG-derived power in the alpha frequency range allows the discrimination between different states of arousal. To calculate alpha power, we adopted spatio-spectral decomposition (SSD; *Nikulin et al., 2011*) which extracts oscillatory sources from a set of mixed signals. Based on generalized eigenvalue decomposition, it finds the linear filters that maximize the signal in a specific frequency band and minimize noise in neighbouring frequency bands. Preprocessing with SSD has been previously shown to increase classification accuracy in BCI applications (*Haufe et al., 2014a*). The alpha frequency range is typically fixed between 8 and 13 Hz. The individual alpha peak frequency, however, varies intra- and inter-individually, for example, with age or cognitive demand (*Haegens et al., 2014*; *Mierau et al., 2017*). To detect each participant's individual peak of alpha oscillations for the SSD, (1) the power spectral density (PSD) of each channel was calculated using Welch's method (*segment length* = 5s * *sampling frequency* [i.e., 250 Hz] with 50 % overlap) in MATLAB (*pwelch* function). (2) To disentangle the power contribution of the $1/f$ aperiodic signal from the periodic component of interest (i.e., alpha), the MATLAB wrapper of the FOOOF toolbox (v0.1.1; *Haller et al., 2018*; frequency range: 0–40 Hz, peak width range: 1–12 Hz, no minimum peak amplitude, threshold of two SDs above the noise of the flattened spectrum) was used. The maximum power value in the 8–13 Hz range was considered the individual alpha peak frequency $\alpha_i$, on which the SSD bands of interest were defined (bandpass signal $\alpha_i \pm 2$ Hz, bandstop noise $\alpha_i \pm 3$ Hz, bandpass noise $\alpha_i \pm 4$ Hz).

The entire procedure was separately applied to the nomov and the mov condition to account for potential peak variability (*Haegens et al., 2014*; *Mierau et al., 2017*). SSD was then computed based on these peaks. A summary of the resulting individual alpha peak frequencies can be found in *Figure 2—source data 1*. *Figure 2* shows the averaged power spectrum across all participants and electrodes. A clearly defined peak in the alpha frequency range is discernible for both conditions (nomov, mov) as well as for states of high and low emotional arousal.

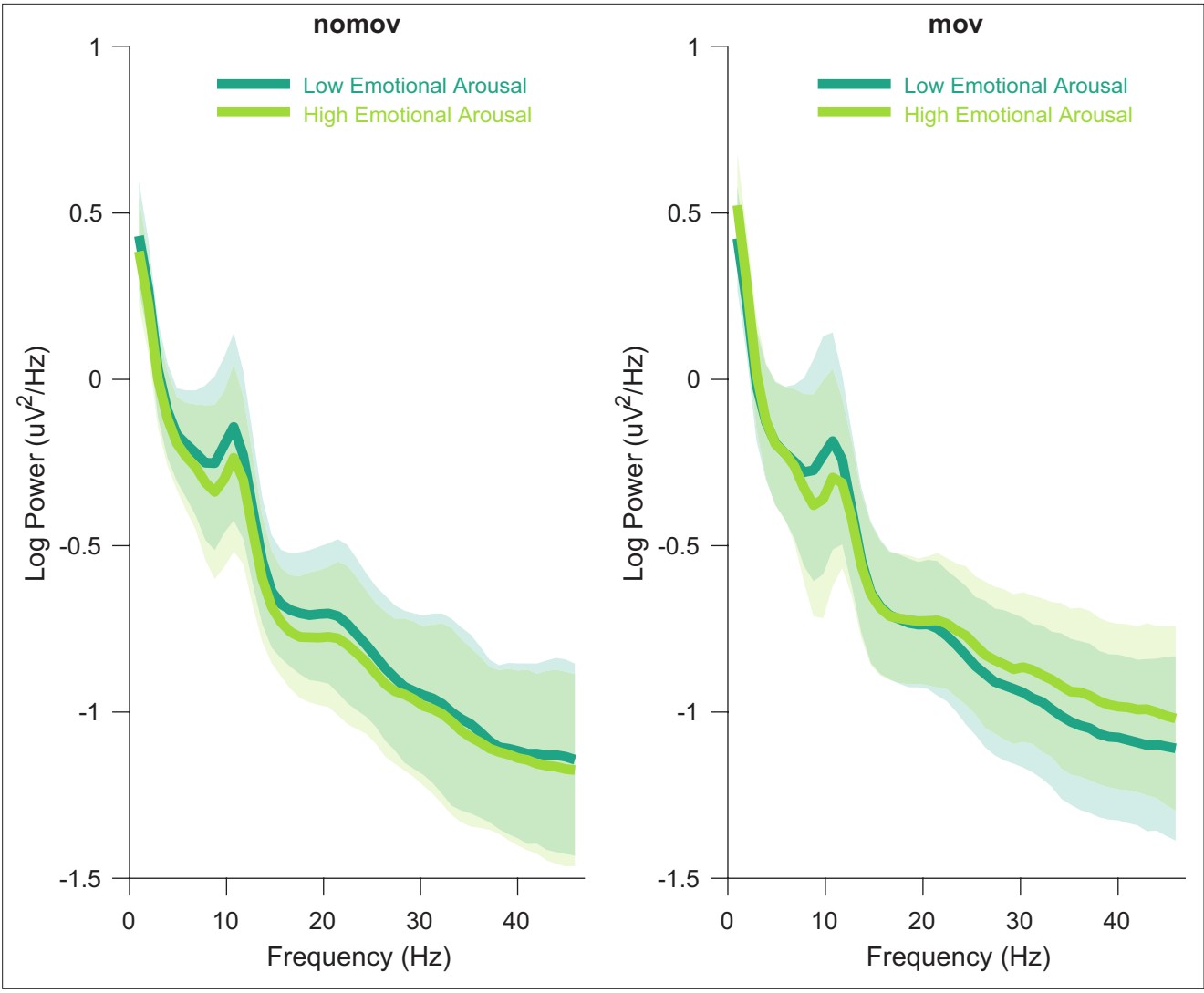

**Figure 2.** Group averaged power spectra for the two emotional arousal levels (low, high) and head movement conditions (nomov, mov). Thick lines represent the mean log-transformed power spectral density of all participants and electrodes. Shaded areas indicate the standard deviation of the participants. High and low emotional arousal are moments that have been rated as most (top tertile) and least arousing (bottom tertile), respectively (the middle tertile was discarded; see main text). The power spectra were produced using MATLAB's *pwelch* function with the same data (after ICA correction and before spatio-spectral decomposition [SSD] filtering) and parameters as the individual alpha peak detection (see Materials and methods section for details). A tabular overview of the alpha peak frequencies of the individual participants is available as *Figure 2—source data 1*.

The online version of this article includes the following figure supplement(s) for figure 2:

**Source data 1.** Selected alpha peaks (8–13 Hz) per participant and condition.

## SSD components selection

The SSD components with sufficient alpha information (i.e., power in the alpha frequency range that exceeds the noise level) were selected with the following steps (see *Figure 3*):

1. The PSD of a component was calculated using Welch's method (*segment length = 5s ∗ sampling frequency* [i.e., 250 Hz] with 50 % overlap) implemented in SciPy (RRID:SCR_008058, v1.4.1; *Jones et al., 2001*).
2. The 1 /*f* curve was fitted in the signal range between 0 and 40 Hz, excluding a ± 4 Hz window around the individual *alpha peak frequency α*i of the subject *i*. The 1 /*f* curve was defined (in log scale) as $f^{-1} = log\left(\frac{1}{a \cdot x^b}\right)$, where x is the given component in the frequency domain, *a* serves as stretch parameter, and *b* represents the slope of the 1 /*f* curve.

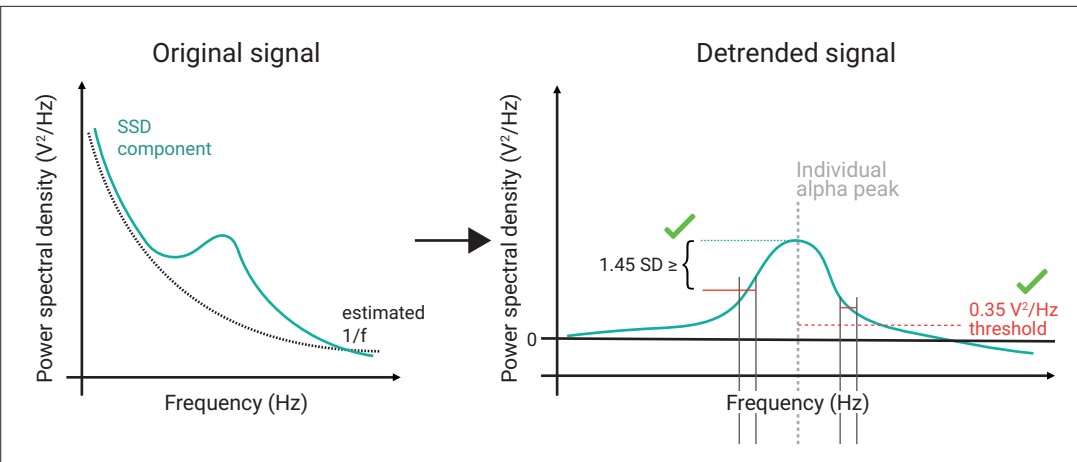

**Figure 3.** Schematic of the selection of individual alpha components using spatio-spectral decomposition (SSD). (*Left*) 1/*f* estimation (dotted grey line) to detrend SSD components (solid turquoise line). (*Right*) After detrending the signal, components were selected, whose peak in the detrended alpha window (centred at the individual alpha peak, vertical dotted grey line) was (**A**) > 0.35 V²/Hz (indicated by horizontal dotted red line) and (**B**) higher than the bigger of the two mean amplitudes of the adjacent frequency flanks (2 Hz width).

3. After fitting these parameters, the signal was detrended with respect to the estimated 1/*f* curve.

4. Those components were selected, whose alpha peak power in the detrended alpha window (as defined in (1)) was (A) greater than zero plus a decision threshold, which was set to $0.35 \frac{\mu V^2}{Hz}$, and (B) higher than the mean amplitude of the adjacent frequency flanks of 2 Hz width on both sides of the window, that is, $power\,(alphapeak) - (meanpower\,(flank)) \geq 1.45SD$ (after *z*-scoring the detrended signal). The two criteria guaranteed the selection of components with a clearly defined alpha-amplitude peak over the noise-threshold defined by $f^{-1}$ (see **Figure 3**).

Particularly the combination of SSD with narrow-band filtering in the alpha-frequency range lowers the probability of signal contamination elicited by artefact-related oscillations, which are typically strongest in frequency ranges above (e.g., muscular activity; **Muthukumaraswamy, 2013**) or below the alpha band (e.g., skin potentials, **Kappenman and Luck, 2010**, or eye blinks, **Manoilov, 2007**; for a comprehensive overview, see also **Chaumon et al., 2015**). Decoding models (SPoC, CSP, LSTM; described below) were trained on those subjects with at least four selected SSD components (26 in the nomov and 19 in the mov). On average, 7.63 of 18.81 (40.53 %) in the nomov and 5.63 of 15.22 (36.98 %) SSD components were selected in the mov condition. Importantly, SSD components had spatial topographies corresponding to occipito-parietal and fronto-central source locations, thus covering brain areas previously implicated in emotional arousal and its regulation.

## Source power comodulation

To test the hypothesis that alpha power in the EEG negatively covaries with the continuous level of subjective emotional arousal, SPoC (as described in **Dähne et al., 2014**; NB: throughout the paper, 'SPoC' refers to SPoC$_\lambda$) was applied to EEG data composed of the selected SSD components and filtered around the central individual alpha peak. Formally, SPoC is an extension of CSP (see below) for regression-like decoding of a continuous target variable. The information contained in the target variable is used to guide the decomposition of neural components that is correlated or anti-correlated with it (**Dähne et al., 2014**). SPoC has been shown to outperform more conventional approaches to relate neural time series to continuous behavioural variables (e.g., correlating power extracted in sensor space and/or after blind source separation methods), which also suffer from additional draw-backs (e.g., lack of patterns' interpretability and lack of adherence to the M/EEG generative model; for details, see **Dähne et al., 2014**). The supervised decomposition procedure takes the variable *z* as target, which comprises the continuous arousal ratings (normalized and mean-centred; 270 s per participant). To reach the same temporal resolution as *z* (i.e., 1 Hz), EEG data were epoched into 270 consecutive segments of 1 s length. For a specific epoch (*e*), the power of an SPoC component

($\hat{s} = W^T X$, where $W^T$ corresponds the transpose of the unmixing matrix $W$ and $X$ to the data matrix of $e$ in SSD space) can be approximated by the variance of its signal within that time interval ($Var[\hat{s}](e)$; **Dähne et al., 2014**). SPoC was separately applied to each participant, producing a number of components equal to the number of previously selected SSD components. The stability and significance of the extracted components was tested with a permutation approach (1000 iterations): $z$ values were shuffled to create a surrogate target variable with randomized phase but same auto-correlation (**Theiler et al., 1992**; adapted from the original SPoC function: https://github.com/svendaehne/matlab_SPoC/blob/master/SPoC/spoc.m; **Dähne, 2015**, **Dähne et al., 2014**). In accordance with the primary objective of SPoC to reconstruct the target variable $z$, lambda values ($\lambda$, i.e., optimization criterion of SPoC: component-wise covariance between $z$ and alpha power; sorted from most positive to most negative) and corresponding Pearson correlation values ($r$) between $z$ and the estimated $z_{est}$ (obtained via $z_{est} = Var\left[W^{(i)T}X\right](e)$) were then calculated for each iteration to generate a naive probability density function (i.e., null hypothesis distribution) and to estimate the probability that the correlation value that was calculated with the original target variable $z$ was obtained by chance. Of note, $z_{est}$ denotes the power time course of the spatially filtered signal that maximally covaries with the behavioural variable $z$. Depending on $i$ (i.e., from which side of the SPoC unmixing matrix the component is chosen), $z_{est}$ will be (maximally) positively (left side of the matrix) or (maximally) negatively (right side of the matrix) correlated with $z$. Given our main hypothesis of an inverse relationship between alpha power and self-reported emotional arousal, we therefore only retained, for each participant, the component with the most negative (precisely: 'smallest') lambda value $\lambda$ (disregarding the p-value to avoid circularity; **Kriegeskorte et al., 2009**), corresponding to the last column of the unmixing matrix $W$.

In line with our hypothesis, single participants' p-values were then obtained by computing the number of permuted $r$ values that were smaller than the one estimated with SPoC.

Crucially, since the extracted linear spatial filters $W$ cannot be directly interpreted (**Haufe et al., 2014b**), topographical scalp projection of the components are represented by the columns of the spatial patterns matrix $A$ obtained by inverting the full matrix $W$ (Figure 6).

## Common spatial patterns

To further test the hypothesis of a link between alpha power and subjective emotional arousal, we aimed to distinguish between the most and the least arousing phases of the experience by using features of the alpha bandpower of the concurrently acquired EEG signal. We followed an approach which has successfully been used in BCI research to discriminate between event- or state-related changes in the bandpower of specific frequency ranges in the EEG signal: the common spatial patterns algorithm specifies, by means of a generalized eigenvalue decomposition, a set of spatial filters to project the EEG data onto components whose bandpower maximally relates to the prevalence of one of two dichotomous states (**Blankertz et al., 2008**; **Ramoser et al., 2000**). In our case, we were interested in distinguishing moments that had been rated to be most (top tertile) and least arousing (bottom tertile).

Using the EEGLAB extension BCILAB (RRID:SCR_007013, v1.4-devel; **Kothe and Makeig, 2013**), data of the selected SSD components, bandpass filtered around the individual alpha peak ±2 Hz, were epoched in 1 s segments. This sample length was chosen to enable the extraction of neural features and corresponding changes in the subjective experience, while maximizing the number of samples from the sparse datasets. Epochs with mid-level arousal ratings (middle tertile) were discarded, yielding 180 epochs (90 per class) for each subject (per movement condition). To assess the classification performance, a randomized 10-fold cross-validation procedure, a common solution for sparse training data (**Bishop, 2006**), was used. Per fold, a CSP-based feature model was calculated on the training data by decomposing the signal of the selected SSD components according to the CSP algorithm. A feature vector comprising the logarithmized variance of the four most discriminative CSP components (using two columns from each side of the eigenvalue decomposition matrix as spatial filters) was extracted per epoch. Data from the training splits were used to train a linear discriminant analysis (LDA) on these feature vectors (**Fisher, 1936**). Covariance matrices used for calculating the LDA were regularized by applying the analytic solution implemented in BCILAB (**Ledoit and Wolf, 2004**). The LDA model was then used to classify the feature vectors extracted from the epochs in the test split to predict the according arousal label. Average classification accuracy

(defined as $1 - \textit{misclassification rate}$) over the 10 folds was taken as the outcome measure to assess the predictive quality of the approach. To allow a spatial interpretation of the projections, like with the SPoC components, the spatial patterns of the two most extreme CSP components (associated with the largest and smallest eigenvalue) that were used to calculate the feature vectors for the linear classification were plotted in Figure 6 (normalized and averaged across subjects per condition) and *Figure 6—figure supplement 1* (per single subject and condition). Source localized patterns are shown in Figure 9.

## Sub-blocked cross-validation

For non-stationary, auto-correlated time-series data, randomized cross-validation can inflate the decoding performance (*Roberts et al., 2017*). To assess and minimize this possibility, we tested whether a blocked cross-validation, which preserves temporal neighbourhood structures among samples, changes the classification results of the CSP analysis. To ensure balanced classes in the training set, the 'synthetic minority oversampling technique', which oversamples the less frequently represented class, was applied (*Chawla et al., 2002*; as implemented in *Larsen, 2021*). The test set was left unbalanced as oversampling of test data can invalidate the assessment of model performance (*Altini, 2015*), and the area under the curve of the receiver operating characteristic (ROC-AUC) was used as a performance measure. To avoid homogeneous test sets (i.e., with samples from only one target class), which (1) would occur in many subjects after 'conventional' chronological cross-validation and (2) would preclude ROC-AUC calculation, a 'sub-blocked' cross-validation was used: for each subject, the dataset was split into three sub-blocks of equal length, which were then used to stratify the data assignment for a (sub-blocked) chronological 10-fold cross-validation. In this design, each fold consists of a concatenation of equally sized stretches of consecutive data samples taken from each of the sub-blocks: for example, to build the validation set in the first fold $[x_1, x_2, x_3]$, with $x_i$ being the $n$ first samples from the $i$th sub-block where $n$ is the total number of samples in the dataset divided by 10 * 3 (number of folds * number of sub-blocks). Thereby the temporal neighbourhood structure among data samples is largely preserved when splitting them into training and testing sets. The (smaller) test set is still sampled from different parts of the experience, which decreases the risk of obtaining homogeneous test sets (e.g., only 'low arousing' sections).

## Source localization

Exact low-resolution tomography analysis (eLORETA, RRID:SCR_007077; *Pascual-Marqui, 2007*) was used to localize the sources corresponding to the component extracted via SPoC and CSP. Our pipeline was based on the work of *Idaji et al., 2020*, who customized the eLORETA implementation of the M/EEG Toolbox of Hamburg (https://www.nitrc.org/projects/meth/).

Our forward model was constructed via the New York Head model (*Haufe et al., 2014b*; *Haufe and Ewald, 2019*; *Huang et al., 2016*) with approximately 2000 voxels and by using 28 out of 30 scalp electrodes (TP9 and TP10 were removed because they are not contained in the model). Crucially, we focused on dipoles perpendicular to the cortex. eLORETA was then used to construct a spatial filter for each voxel from the leadfield matrix, and respective sources were computed by multiplying the resultant demixing matrix with the spatial patterns $A$ of the selected SPoC and CSP components. Inverse modelling was computed separately per participant and condition before it was averaged for each condition across all subjects (Figure 9).

## LSTM RNN

Deep learning models have become a useful tool to decode neural information (e.g., *Agrawal et al., 2014*; *Khaligh-Razavi and Kriegeskorte, 2014*). Applying a deep learning approach to the time series of EEG recordings (e.g., *Bashivan et al., 2016*) can be achieved using LSTM RNNs (*Hochreiter and Schmidhuber, 1995*; *Hochreiter and Schmidhuber, 1997*). With their property to store and control relevant information over time, they can find adjacent as well as distant patterns in (time) sequential data. The LSTM analysis was implemented in the Python-based (RRID:SCR_008394) deep learning library *TensorFlow* (RRID:SCR_016345, v1.14.0; Google Inc, USA; *Abadi et al., 2015*; *Zaremba et al., 2015*).

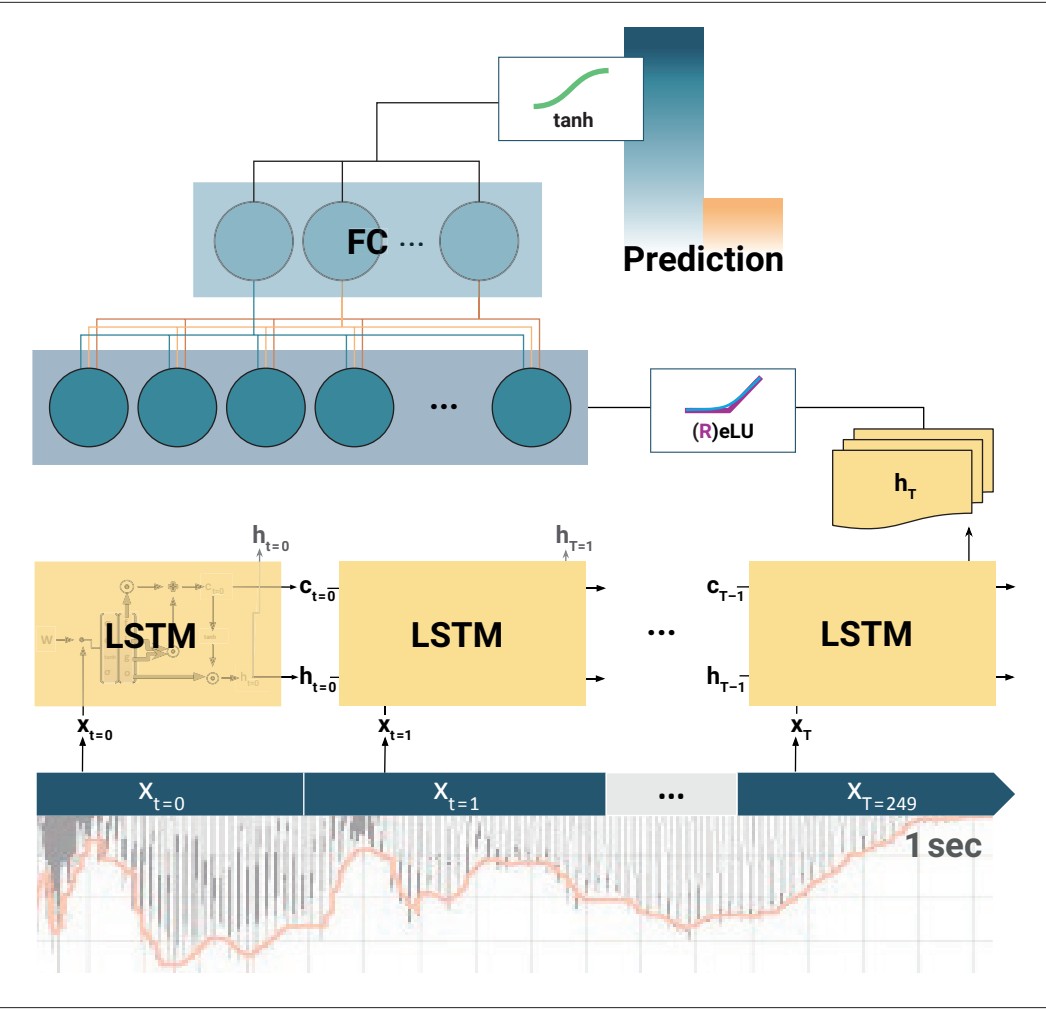

**Figure 4.** Schematic of the long short-term memory (LSTM) recurrent neural network (RNN). At each training step, the LSTM cells successively slide over 250 data arrays of neural components ($x_{t=0}$, $x_{t=1}$,..., $x_{T=249}$) corresponding to 1 s of the EEG recording. At each step $t$, the LSTM cell computes its hidden state $h_t$. Only the final LSTM output ($h_T$) at time-step $T = 249$ is then fed into the following fully connected (FC) layer. The outputs of all (LSTMs, FCs) but the final layer are normalized by rectified linear units (ReLU) or exponential linear units (ELU). Finally, the model prediction is extracted from the last FC layer via a tangens hyperbolicus (*tanh*). Note: depending on model architecture, there were one to two LSTM layers, and one to two FC layers. The hyperparameter constellations that yielded the highest accuracy for the individual participants per movement condition are available as *Figure 4— source data 1*.

The online version of this article includes the following figure supplement(s) for figure 4:

**Source data 1.** Long short-term memory (LSTM) hyperparameter search per movement condition.

## Model architecture and hyperparameter search

Deep learning models usually have a high variance due to random weight initialization, architectural choices, and hyperparameters (HPs; *Geman et al., 1992*; but see *Neal et al., 2019*). We here used a two-step random search (*Bergstra and Bengio, 2012*) strategy in order to find optimal HPs, to reduce the model variance and make the search computationally feasible. First, a broad random search was applied on a random subset of 10 subjects (20 random combinations) in each condition. Then, the two best HPs per subject were taken and applied to the datasets of all subjects. Due to time constraints and computational complexity, the HP search was limited to a predefined range of settings and the model architecture was constrained to maximal two LSTM layers followed by maximal two fully connected layers (FC; *Hefron et al., 2017*; see *Figure 4*). Each layer size $ls_l$ varied between 10 and 100 nodes ($ls_l$ ∈ 10, 15, 20, 25, 30, 40, 50, 65, 80, 100), and a successive layer needed to be equal or smaller in size

(bottleneck architecture). The output of each layer was squashed through either rectified linear units or exponential linear units, which both allow for steeper learning curves in contrast to conventional activation functions such as sigmoid nonlinearity (*Clevert et al., 2016*). The output of the last network layer ($FC_L$) was fed into a tangens hyperbolicus (*tanh*) to match the binned ratings, which were labelled with –1 or 1, respectively. We applied a mean-squared error loss to calculate the difference between the model output (i.e., the prediction) and the labels, leading to a stronger weighting of losses at the upper- or lower-class border, respectively. To control and tax too large model weights, different regularization methods (*L1, L2*) with different regularization strengths ($\Lambda \in 0.00, 0.18, 0.36, 0.72, 1.44$) were tested. Weights were optimized using *Adam* (learning rate: $lr \in 1e^{-2}, 1e^{-3}, 5e^{-4}$) due to its fast convergence (*Marti, 2015*; see also *Ruder, 2017*). The number of input components (SSD, $N_{comp}$: $N \in$ [1, 10]) was treated as HP. The specific $N_{comp}$ neural components were successively drawn according to the order of the SSD selection.

## Training procedure

The final dataset per subject was a three-dimensional tensor of size 270 × 250 × 10 *(epochs × samples × components)*. If less than 10 components were extracted for a given subject, the tensor was filled with zero vectors. After some test runs and visual observation of the convergence behaviour of the learning progress, training iterations were set to 20 (i.e., the model ran 20 times through the whole training dataset). The 1 s samples were fed into the LSTM in random mini-batches of size 9 (bs = 9), since training on batches allows for faster and more robust feature learning (*Ruder, 2017*), leading to the following input tensor at training step *ts*: $x_{train,ts}^{bsx250x10}$ .

## Statistical evaluation

To test whether the results of the binary modelling approaches (CSP, LSTM) were statistically significantly above chance level, exact binomial tests were conducted per subject and experimental condition (nomov, mov) over all 180 epochs of the respective time series (nomov, mov). To do so, for each of the binary modelling approaches (CSP, LSTM), the predictions for the single epochs in the 10 test splits of the cross-validation were concatenated to a single vector. The proportion of correct and false predictions was then compared to a null model with prediction accuracy 0.5 (chance level). To test the average (across subjects) classification accuracies of the binary models, we calculated one-sided one-sample *t*-tests, comparing the mean accuracy of the respective model for both experimental conditions against the theoretical prediction accuracy of a random classifier (0.5). To test whether classification accuracies differed between the two models (CSP, LSTM) or between the experimental conditions (nomov, mov), a repeated-measures two-way ANOVA was conducted on the accuracy scores of all subjects with preprocessed data from both conditions (*n* = 18).

To account for potential biases due to auto-correlations in the time series which might affect the statistical evaluation of the classification model, in an additional control analysis, block permutation testing was applied to the CSP approach: to maintain a local auto-correlative structure similar to the original data in the permuted target vectors, the time series were split into 10 equally sized blocks, which were then shuffled while the internal temporal structure of each block remained intact (*Winkler et al., 2014*). To test whether the actual decoding scores (from non-permuted data) were significantly above chance level, we assessed their percentile rank in relation to the null distributions (1000 permutations) on the single-subject level. On the group level, one-sided paired *t*-tests were used to compare the distribution of the actual decoding results against the distribution of the means of the null distributions per subject. Due to its high computational processing cost and duration, we did not perform permutation testing for the LSTM model.

For SPoC, in addition to the aforementioned within-participants permutation approach yielding a single p-value for each component, group-level statistics were assessed: the hypothesis of a negative correlation between alpha power and emotional arousal was tested with a one-sample, one-tailed *t*-test on the correlation values between *z* and $z_{est}$, which assessed whether the mean correlation value per condition was significantly lower than the average of the permuted ones.

The code for preprocessing of the data, the three prediction models, and the statistical evaluation is available on GitHub (https://github.com/NeVRo-study/NeVRo; *Hofmann et al., 2021*; copy archived at swh:1:rev:669d5c2d6c73cbb70422efb933916fe8304195b5).

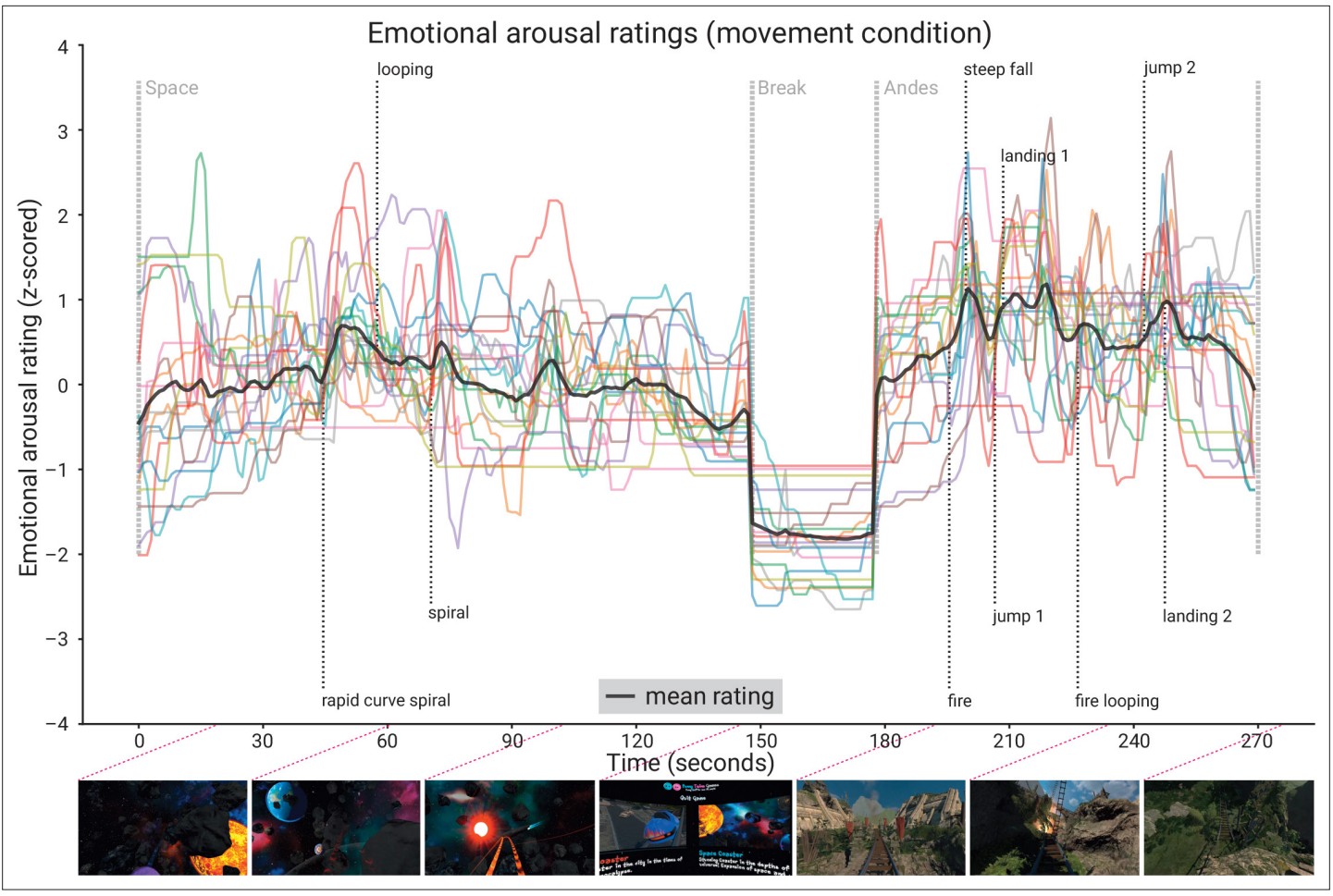

**Figure 5.** Subjective emotional arousal ratings (movement condition). Emotional arousal ratings of the experience (with head movement; see *Figure 5—figure supplement 1* for the ratings from the no-movement condition). Coloured lines: individual participants; black line: mean across participants; vertical lines (light grey): beginning of the three phases (Space Coaster, Break, Andes Coaster); vertical lines (dark grey): manually labelled salient events (for illustration). Bottom row: exemplary screenshots of the virtual reality (VR) experience. The ratings for the condition without head movement are shown in the figure supplement.

The online version of this article includes the following figure supplement(s) for figure 5:

**Figure supplement 1.** Subjective emotional arousal ratings (no-movement condition).

## Control analysis: excluding the break for model training

The 30 s break differed from the rollercoaster rides in visual features (e.g., static vs. dynamic input) and in arousal ratings, which were constantly relatively low during the break (see *Figure 5*). Thus, the break contributed mainly to the 'low arousing' class. To test whether the decoding approaches also succeed if the break section is excluded from the analysis, SPoC and CSP decoding were repeated for the data without the break, that is, the rollercoasters only (240 s in total). The LSTM approach was skipped in this control analysis due to its computational processing cost and duration, and the comparable performance with CSP in the main analysis. For the classification (CSP), the tertile split on the subjective arousal ratings was recalculated such that the class of 'low arousal' segments now comprises the least arousing sections of the rollercoasters. We then trained and tested the SPoC and CSP models with the procedures that were used for the original dataset (incl. the break). For maximal stringency, we used the sub-blocked cross-validation and block permutation approach to assess the performance of the CSP model. To test whether excluding the break changed the model performance, we compared the distributions of the decoding performance parameters (SPoC: Pearson correlation

with target; CSP: ROC-AUC) from the data with and without the break using two-sided paired *t*-tests. We did this per model and movement condition.

## Results

### Participants

Forty-five healthy young participants (22 men, M ± SD: 24.6 ± 3.1, range: 20–32 years) completed the experiment. Data from eight participants needed to be discarded due to technical problems (*n* = 5) or electrode malfunctioning (*n* = 1); one participant discontinued the experiment and another participant reported having taken psychoactive medication. The data from 37 participants entered the analysis (17 men, age: M ± SD: 25.1 ± 3.1, range: 20–31 years). After quality assurance during the EEG preprocessing, data from 26 participants in the condition with no head movement (nomov) and 19 in condition with free head movement (mov) entered the statistical analyses that included EEG data.

### Self-reports

#### Questionnaires

From before (M ± SD: 8.68 ± 2.82, range: 6–17) to after the experiment (M ± SD: 11.82 ± 5.24, range: 6–29), the overall simulator sickness (e.g., nausea, headache) increased significantly ($t(36) = 3.72$, p = 0.0007). As the trait questionnaires are not the focus of this study, their results will be reported elsewhere.

#### Emotional arousal ratings

The retrospective emotional arousal ratings for the VR experience, averaged across all subjects and timepoints, were 46.94 ± 12.50 (M ± SD, range: 16.17–66.29) in the nomov and 50.06 ± 12.55 (M ± SD, range: 18.00–69.94) in the mov condition. Qualitatively, the emotional arousal was highest for the *Andes Coaster*, lower for the *Space Coaster*, and lowest for the break (see *Figure 5*).

### Neurophysiology

#### SPoC

SPoC results showed that for 24/26 (92.30%) participants in the nomov and 16/19 (84.21%) participants in the mov condition (see *Figure 10—source data 1* for single-participant results), the component with the highest absolute lambda value corresponded to the one that maximized the negative correlation between *z* (normalized and mean-centred subjective ratings) and alpha power. Based on permutation-based testing (1000 iterations; exact p values are reported in *Figure 10—source data 1*), the negative correlation was statistically significant (p < 0.05) in 8/26 (30.76%) participants for the nomov and 7/19 (36.84%) participants for the mov condition. The global mean lambda value of these components was –0.46 for the nomov (range: –1.49 to +0.08) and –0.42 for the mov condition (range: –1.49 to +0.02). The mean Pearson correlation value between the target variable *z* and $z_{est}$ (estimated target variable) was significantly lower than the average of single participants' permuted ones for both the nomov (M ± SD: –0.25 ± 0.12; range: –0.53 to + 0.09; $t_{nomov}(25) = -3.62$; p < 0.01) and the mov condition (M ± SD: –0.25 ± 0.12; range: –0.52 to +0.04; $t_{mov}(18) = -3.13$; p < 0.01).

#### CSP

The classifier based on CSP was able to decode significantly above chance level whether a subject experienced high or low emotional arousal during a given second of the experience. On average, the classification accuracy was 60.83% ± 7.40 % (M ± SD; range: 47.22–77.78%) for the nomov, and 60.76% ± 6.58 % (M ± SD; range: 48.33–71.67%) for the mov condition. Both were significantly above chance level ($t_{nomov}(25) = 7.47$, $p_{nomov} < 0.001$; $t_{mov}(18) = 7.12$, $p_{mov} < 0.001$). At the single-subject level, the classification accuracy was significantly above chance level (p < 0.05) for 17/26 (65.38%) participants in the nomov, and for 12/19 (63.16%) participants in the mov condition (see *Figure 10—source data 1* for single-participant results). The spatial patterns yielded by the CSP decomposition are shown in *Figure 6* (across participants) and in *Figure 6—figure supplement 1* (individual participants). Corresponding alpha power sources (located via eLORETA) are shown in Figure 9.

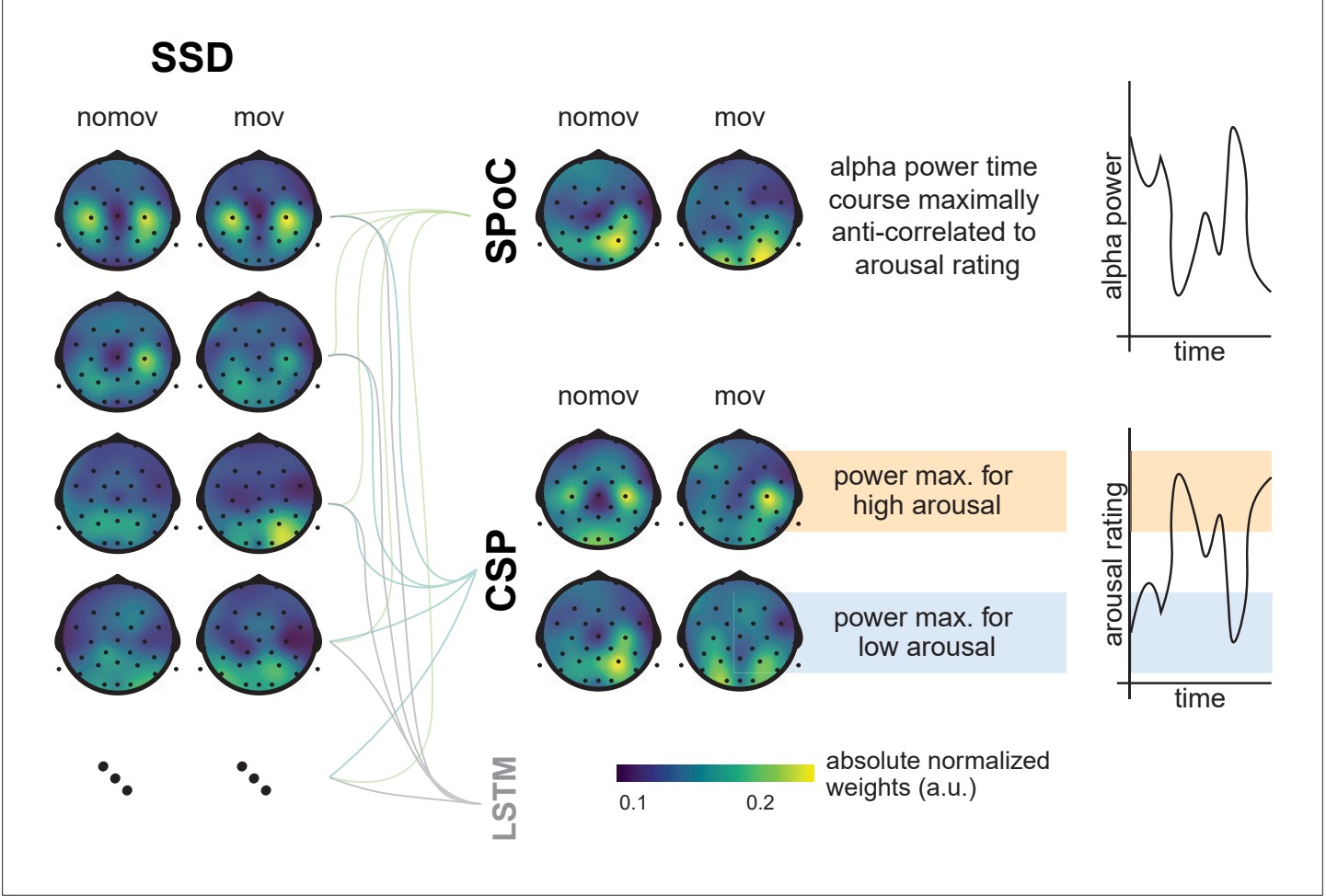

**Figure 6.** Spatial patterns resulting from spatio-spectral decomposition (SSD), source power comodulation (SPoC), and common spatial pattern (CSP) decomposition. Colours represent absolute normalized pattern weights (inverse filter matrices) averaged across all subjects per condition (nomov: without head movement, mov: with head movement). Before averaging, the pattern weight vectors of each individual subject were normalized by their respective L2-norm. To avoid cancellation due to the non-polarity-aligned nature of the dipolar sources across subjects, the average was calculated from the absolute pattern weights. SSD allows the extraction of components with a clearly defined spectral peak in the alpha frequency band. Shown are the patterns associated with the four SSD components that yielded the best signal-to-noise ratio (*left column*). The SSD filtered signal was the input for the decoding approaches SPoC, CSP, and LSTM: SPoC adds a spatial filter, optimizing the covariance between the continuous emotional arousal ratings and alpha power. Shown here is the pattern of the component which – in line with our hypothesis – maximized the inverse relationship between emotional arousal and alpha power. CSP decomposition yielded components with maximal alpha power for low-arousing epochs and minimal for high-arousing epochs (*bottom row in the CSP panel*) or vice versa (*upper row in the CSP panel*). The high correspondence between the patterns resulting from SPoC and CSP seems to reflect that both algorithms converge to similar solutions, capturing alpha power modulations in parieto-occipital regions as a function of emotional arousal. The spatial patterns for the individual subjects are displayed in the figure supplement. (Note: as the LSTM results cannot be topographically interpreted, they are not depicted here.)

The online version of this article includes the following figure supplement(s) for figure 6:

**Figure supplement 1.** Spatial patterns per single subject and movement condition yielded by the different spatial signal decompositions.

To test for potential biases from the model or the data, specifically its auto-correlative properties, we ran a control analysis for CSP using sub-blocked chronological cross-validation and block permutation for statistical evaluation on the single-subject level.

Also under these – more strict – evaluation criteria, the average decoding performance (ROC-AUC) for CSP was significantly above chance level, both in the nomov (ROC-AUC: 0.61 ± 0.09 M ± SD, range: 0.42–0.79; $t(25) = 4.59$, $p < 0.001$) and in the mov condition (ROC-AUC: 0.60 ± 0.09 M ± SD, range: 0.44–0.74; $t(18) = 3.27$, $p < 0.01$). On the single-subject level (as assessed by permutation

tests), decoding performance was significantly ($p < 0.05$) higher when decoding the actual, unpermuted labels compared to the block-permuted labels for 9/26 (34.62%) participants in the nomov and 5/19 (26.32%) participants in the mov condition.

## LSTM

After a random search over a constrained range of HPs, we extracted the best individual HP set per subject (see *Figure 4—source data 1* for the list of best HPs per condition). The mean classification accuracy was 59.42% ± 4.57 % (M ± SD; range: 52.22–68.33%) for the nomov, and 61.29% ± 4.5 % (M ± SD; range: 53.89–71.11%) for the mov condition. Both were significantly above chance level ($t_{nomov}(25) = 10.82$, $p_{nomov} < 0.001$; $t_{mov}(16) = 10.51$, $p_{mov} < 0.001$). At the single-subject level, the classification accuracy was significantly above chance level for 16/26 (61.54%) participants in the nomov condition, and for 16/19 (84.21%) participants in the mov condition (same test as for CSP results; see *Figure 10—source data 1*).

## Comparison of model performances

As an illustration of the prediction behaviour across all three models in one participant (with high performance for all three decoding approaches), see *Figure 7*. Correlations of performances across models and experimental conditions are shown in Figure 10. The (positive) correlation between the two binary classification approaches (CSP, LSTM) was significant (after Bonferroni multiple-comparison correction), irrespective of the experimental condition (nomov, mov), meaning that subjects who could be better classified with CSP also yielded better results in the LSTM-based classification. In a repeated-measures ANOVA testing for differences in the accuracies of the two binary classification models (CSP, LSTM) and the two conditions (nomov, mov), none of the effects was significant: neither the main effect *model* ($F(1,17) = 0.02$, $p = 0.904$) nor the main effect *condition* ($F(1,17) = 0.72$, $p = 0.408$) or their interaction ($F(1,17) = 1.59$, $p = 0.225$). For a further comparison of the performances of the classification approaches, the respective confusion matrices are depicted in *Figure 8* (average across the subjects per condition and model).

## Control analysis: excluding the break from model training

SPoC and CSP performed significantly above chance level also when trained and tested on data without the break section.

For CSP on data without the break, the average classification performance (ROC-AUC) was 0.57 ± 0.10 (M ± SD; range: 0.28–0.78) in the nomov and 0.59 ± 0.09 (M ± SD; range: 0.45–0.77) in the mov condition (see previous paragraph for the decoding performance with the break included). Average model performances were still significantly above chance level (means of the block permutation distributions on the single-subject level) in both movement conditions (nomov: $t(25) = 2.89$, $p < 0.01$; mov: $t(18) = 3.50$, $p < 0.01$). On the single-subject level, the classification performance was significantly above chance level for 3/26 (11.54%) participants in the nomov and 5/19 (26.32%) participants in the mov condition.

For SPoC on data without the break, the average Pearson correlation between $z$ and $z_{est}$ (estimated target variable) was significantly smaller (more negative) than the average of single participants' permuted correlation values for both the nomov (M ± SD: –0.22 ± 0.08; range: –0.36 to –0.07; $t_{nomov}(25) = –3.17$; $p < 0.01$) and the mov condition (M ± SD: –0.21 ± 0.07; range: –0.37 to –0.061; $t_{mov}(18) = –2.53$; $p < 0.05$). On the single-subject level, 2/26 (7.69%) participants for the nomov and 7/19 (36.84%) participants for the mov condition remained statistically significant ($p < 0.05$) after permutation-based tests.

Removing the break from the training data overall numerically decreased the decoding performances of both models. For CSP, the decrease was significant in the nomov ($t(25) = 2.23$, $p = 0.034$) and not significant in the mov condition ($t(18) = 0.57$, $p = 0.58$). For SPoC, the decrease (Pearson correlation) was not significant in both conditions (nomov: $t(25) = –1.66$, $p = 0.108$; mov: $t(18) = –1.13$, $p = 0.269$).

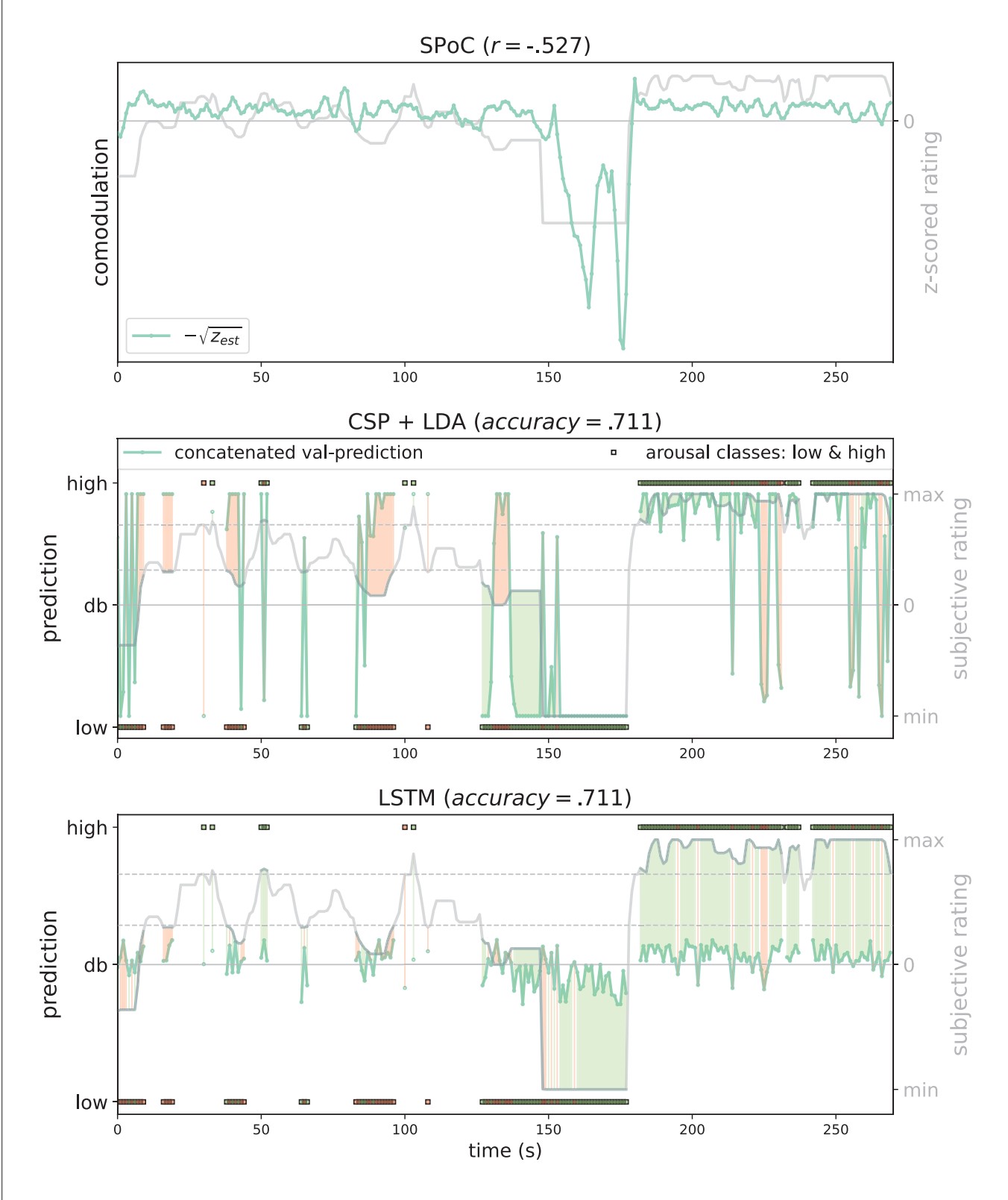

**Figure 7.** Exemplary model predictions. Predictions (turquoise line, dots) across models trained on the data of one participant in the movement condition (source power comodulation [SPoC]: normalized negative $z_{est}$, here comodulation; common spatial patterns [CSP]: posterior probability; long short-term memory [LSTM]: *tanh* output). *Top row*: most negatively correlating SPoC component (for visualization we depict the normalized and mean-centred value of the rating and of the negative square root of $z_{est}$). *Middle and lower row*: model predictions on validation sets (across the

*Figure 7 continued on next page*

*Figure 7 continued*

cross-validation splits) for CSP and LSTM, respectively. The grey curvy line in each panel indicates the continuous subjective rating of the participant. Horizontal dotted lines indicate the class borders. The area between these lines is the mid-tertile which was discarded for CSP and LSTM analyses. Class membership of each rating sample (1 s) is indicated by the circles at the top and bottom of the rating. A model output falling under or above the decision boundary (db) indicates the model prediction for one over the other class, respectively. The correct or incorrect prediction is indicated by the colour of the circle (green and red, respectively), and additionally colour-coded as area between model output (turquoise) and rating.

## Discussion

The general aim of this study was to capture the dynamic relationship between subjective experience and neurophysiology under naturalistic stimulation using immersive VR. The hypothesized link between EEG alpha power and self-reported emotional arousal could be confirmed by relating alpha power components to continuous retrospective ratings of emotional arousal (using SPoC) as well as by decoding states of higher and lower emotional arousal from them (using CSP and LSTMs), particularly in parieto-occipital regions. In addition to extending our knowledge about the functional anatomy of emotional arousal, these findings support previous results from classical studies and confirm them under more naturalistic conditions. They thereby pave the way for real-world scenarios and applications.

### Physiological and psychological concomitants of emotional arousal

In studies with event-related stimulation or block designs, more emotionally arousing compared to less emotionally arousing images, videos, and sounds were associated with event-related decreases in alpha power, predominantly over parieto-occipital electrodes (*De Cesarei and Codispoti, 2011*; *Luft and Bhattacharya, 2015*; *Schubring and Schupp, 2019*; *Uusberg et al., 2013*; *Koelstra et al., 2012*). While such stimuli provide a high degree of experimental control in terms of low-level properties and presentation timings, the emotional experience and its neurophysiology under event-related

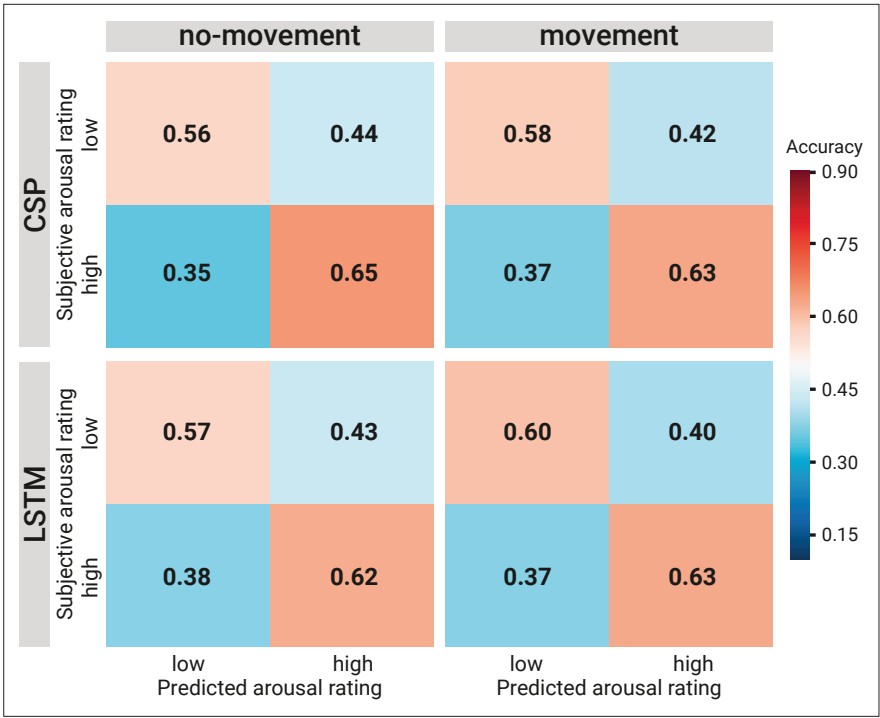

**Figure 8.** Comparison of the binary decoding approaches. Confusion matrices of the classification accuracies for higher and lower self-reported emotional arousal using long short-term memory (LSTM) (*lower row*) and common spatial patterns (CSP) (*upper row*) in the condition without (*left column*) and with (*right column*) head movement. The data underlying this figure can be downloaded as *Figure 8—source data 1*.

The online version of this article includes the following source data for figure 8:

**Source data 1.** Prediction tables of the binary decoding models.

stimulation may differ from the emotional experience in real-life settings, which is perceptually complex, multisensory, and continuously developing over time.

Our results provide evidence that the neural mechanisms reflected in modulations of alpha power – particularly in parieto-occipital regions – also bear information about the subjective emotional state of a person undergoing an immersive and emotionally arousing experience. Also fMRI studies have related brain activity in parietal cortices and emotional processing (e.g., *Lettieri et al., 2019*). Our study thus suggests that findings from event-related, simplified stimulation generalize to more naturalistic (i.e., dynamic and interactive) settings.

Paralleling the idea of emotional arousal being a dimension of 'core affect' (*Russell and Barrett, 1999*) and a psychological primitive that underlies many mental phenomena, also alpha oscillations have been connected to various psychological 'core processes': for instance, modulations of alpha power were linked to attention (*Van Diepen et al., 2019*) and memory (*Klimesch, 2012*). More generally, neural oscillations in the alpha frequency range were suggested to serve functional

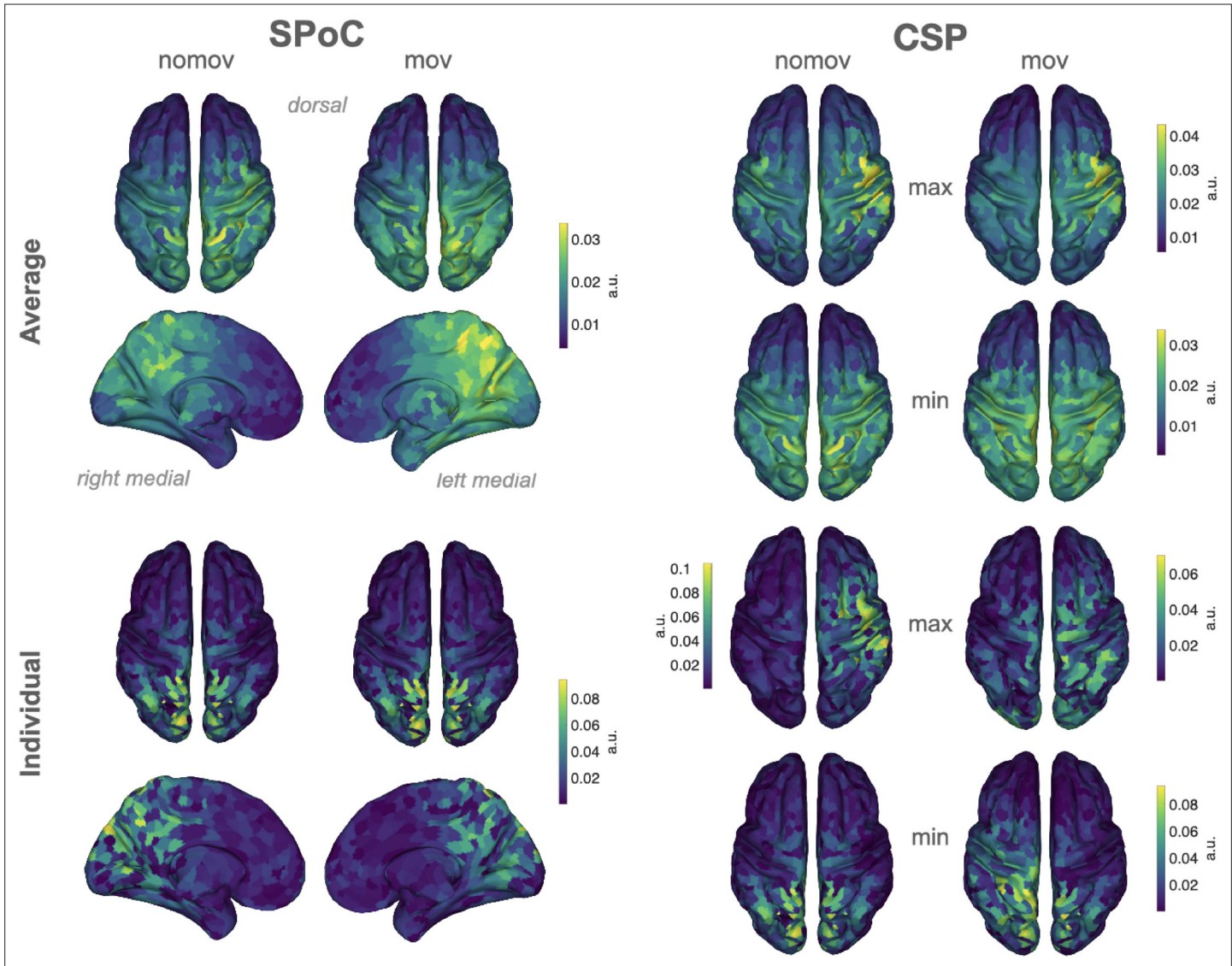

**Figure 9.** Source reconstructions (exact low resolution tomography analysis [eLORETA]). The projection of source power comodulation (SPoC) and common spatial patterns (CSP) components in source space confirms the link between emotional arousal and alpha oscillations in parieto-occipital regions. Colours represent the inversely modelled contribution of the cortical voxels to the respective spatial pattern yielded by SPoC or CSP (max: component maximizing power for epochs of high arousal; min: component minimizing power for epochs of high arousal). We applied the same normalization and averaging procedures as for the topoplots in *Figure 6*. *Upper row*: averaged across all subjects per condition (nomov, mov). *Lower row*: patterns of one individual (the same as in *Figure 7*).

inhibition of irrelevant sensory input (*Jensen and Mazaheri, 2010*; *Foster and Awh, 2019*) and to code for the location and the timing of task-relevant stimuli (*Foster et al., 2017*). Such processes can be functionally linked to emotional arousal: during emotionally arousing experiences, preferred and enhanced processing of relevant sensory stimuli (e.g., indicating potential threats) is an adaptive behaviour. In line with this, modulations of alpha oscillations over parietal sensors have been linked to threat processing (*Grimshaw et al., 2014*). Variations in emotional arousal and alpha power may, thus, have guided attention and memory formation also in our experiment: during particularly arousing parts of the rollercoaster, participants may have directed their attention to specific parts of the visual scene, for example, to foresee the end of the looping. Moreover, our inverse modelling (*Figure 9*) has also localized arousal-related alpha sources in sensorimotor cortices, which could correspond to somatic experiences typically associated with rollercoasters. Some of the averaged spatial patterns (see *Figures 6 and 9*) we observed for the SPoC- and CSP-based decoding stronger absolute weights for electrodes above right – as compared to left – cortices. Since we did not hypothesize a lateralization of the alpha effects, we refrained from statistically testing differences between the hemispheres. Similar patterns of right-lateralized alpha oscillations have also been related to arousal in major depression (*Metzger et al., 2004*; *Stewart et al., 2011*). However, it is unclear to which extent these effects are specific to arousal, as lateralization of alpha power has also been observed in working-memory (*Pavlov and Kotchoubey, 2020*) and resting-state studies (*Ocklenburg et al., 2019*). Our results motivate experimental work that will model the link between emotional arousal and alpha oscillations by systematically varying additional variables (e.g., attention, sensorimotor processing). We argue that studying such relationships in naturalistic settings allows embracing and learning statistical interdependencies that are characteristics of the real world.

## VR as a step towards a real-world neuroscience

More naturalistic experimental stimulation, for example, using immersive VR, allows to test the brain under conditions it was optimized for and thereby improve the discovery of neural features and dynamics (*Gibson, 1978*; *Hasson et al., 2020*). Findings from naturalistic studies can test the real-world relevance of results obtained in highly controlled, abstract laboratory settings (*Matusz et al., 2019*; *Shamay-Tsoory and Mendelsohn, 2019*). Challenges of using VR for more naturalistic research designs are the creation of high-quality VR content, more complex technical setups, and discomfort caused by the immersion into the virtual environment (*Pan and Hamilton, 2018*; *Vasser and Aru, 2020*). Despite the incongruence between VR rollercoaster-induced visual stimulation and vestibular signals, which may lead to motion sickness (*Reason and Brand, 1975*), only one of our participants stopped the experiment because of cybersickness. This low number may result from the relatively short length of the VR experience (net length: <20 min) and the professionally produced VR stimulation. Shorter exposure times (*Rebenitsch and Owen, 2016*) and experiences that elicit stronger feelings of presence have been associated with lower levels of cybersickness (*Weech et al., 2019*).

Combining EEG with VR provides additional challenges: the signal-to-noise ratio (SNR) can be decreased due to mechanical interference of the VR headset with the EEG cap and due to movement artefacts when the participant interacts with the virtual environment (e.g., head rotations). To ensure high data quality, we applied multiple measures to prevent, identify, reject, or correct artefacts in the EEG signal (see Materials and methods section for details). Ultimately, the performance of all three decoding models did not differ significantly for both conditions (nomov, mov). We suggest that, with appropriate quality assurance during data acquisition and analysis (leading to more data rejection/correction for mov than for nomov), EEG can be combined with immersive VR and free head movements. Other studies of mobile brain imaging, even recording outdoors and with full-body movements, came to similar conclusions (*Debener et al., 2012*; *Ehinger et al., 2014*; *Gramann et al., 2011*; *Symeonidou et al., 2018*).

## Evaluating EEG data from naturalistic experiments using complementary methods

Each of the applied decoding approaches allows for different insights and interpretations, but overall, they yield converging results.

## SPoC and CSP

SPoC and CSP share advantages that are common to most spatial filtering methods based on generalized eigenvalue decomposition, namely precise optimization policies, high speed, and interpretability. As dimensionality reduction techniques, they combine data from multiple M/EEG channels to obtain a new signal (component) with a higher SNR (*Lotte et al., 2018*; *Parra et al., 2005*). This aids maximizing the difference in the signal of interest between experimental conditions (*de Cheveigné and Parra, 2014*; *Rivet et al., 2009*) or against signals in the neighbouring frequency ranges (*Nikulin et al., 2011*). The similarity between the two approaches (SPoC, CSP) and their interpretability becomes apparent in the resulting spatial patterns: the normalized and averaged SPoC topoplots and source localizations in both conditions (nomov, mov) resemble the ones extracted via CSP to maximize power for the low-arousal epochs of the experience (*Figures 6 and 9*). SPoC and CSP solve a similar problem here: extracting components whose power is minimal during states of high emotional arousal and maximal during states of low arousal.

This indicates that SPoC and CSP exploited similar spatial informational patterns in the input data. However, the datasets handed to the SPoC and CSP models were not identical. For the CSP analysis, only the upper and lower extreme of the arousal ratings were included (i.e. two-thirds of the data), while epochs with medium arousal ratings (i.e., one-third of the data) were excluded, whereas SPoC was trained on the full continuous data stream. There are two potential explanations for the observation that SPoC and CSP nevertheless yield similar spatial patterns: either the most relevant information was encoded in the most extreme parts of the experience, or there is a truly linear relationship between alpha power and emotional arousal that can be queried on all parts of this spectrum ranging from low to high emotional arousal.

The spatial patterns for the components gained from SSD, SPoC, and CSP exhibit discernible variance between the single subjects (see *Figure 6—figure supplement 1*). This can be, for example, caused by physiological differences (e.g., different shapes of the skull, different cortical folding) or slightly different positioning of the EEG electrodes. The same cortical source might thereby lead to different patterns of scalp EEG in different participants. Spatial filtering procedures inverse this projection and the extracted patterns therefore also vary across subjects. Such inter-individual differences are well known for BCIs, and extensions for CSP have been suggested, which allow for a transfer of features across subjects (e.g., *Cheng et al., 2017*). To emphasize the communalities across individual patterns and indicate the cortical areas that contributed most to decoding results, we report the averaged patterns (*Figure 6*) and the averaged results of the reconstructed cortical sources (*Figure 9*).

To test for confounds or analytic artefacts, for example, due to auto-correlations in the data, we additionally applied 'sub-blocked' cross-validation for model training and block permutation for statistical evaluation. Also under these more strict evaluation conditions, the average decoding performance was significantly above chance level. It is therefore unlikely that the results can be explained solely by dependencies in the data (e.g., auto-correlation) which are not explicitly modelled in the main analysis.

Moreover, to test the impact of the differences between the rollercoasters and the break, for example, regarding visual dynamics and elicited emotional arousal, on the decoding performance, SPoC and CSP analyses were repeated on the data without the break. Again, the average decoding performances decreased compared to the data with the break, but remained significantly above chance level for both head movement conditions. The decrease in decoding performance with the break removed may result from (1) less training data being available and (2) a narrower range of emotional arousal values, more similar classes ('high arousal' and 'low arousal'), and therefore a more difficult distinction.

We observed a high degree of variability in decoding performance across participants (see *Figure 10*). For example, for less than 70 % (and less than 35 % with sub-blocked cross-validation and permutation testing) of participants, CSP yielded significant results on the single-subject level. This variability reflects the difficulty of some features and classifiers to perform equally well across subjects, which has been reported in the BCI literature (*Krusienski et al., 2011*; *Nurse et al., 2015*). In a supplementary analysis, we compared the classification results to a less complex logistic regression model, which was directly trained on time-frequency data from electrodes in the occipital-parietal region of interest. The model performed almost on par with CSP in the mov condition but was less sensitive in the nomov condition. Linear regression on time-frequency data in sensor space also has

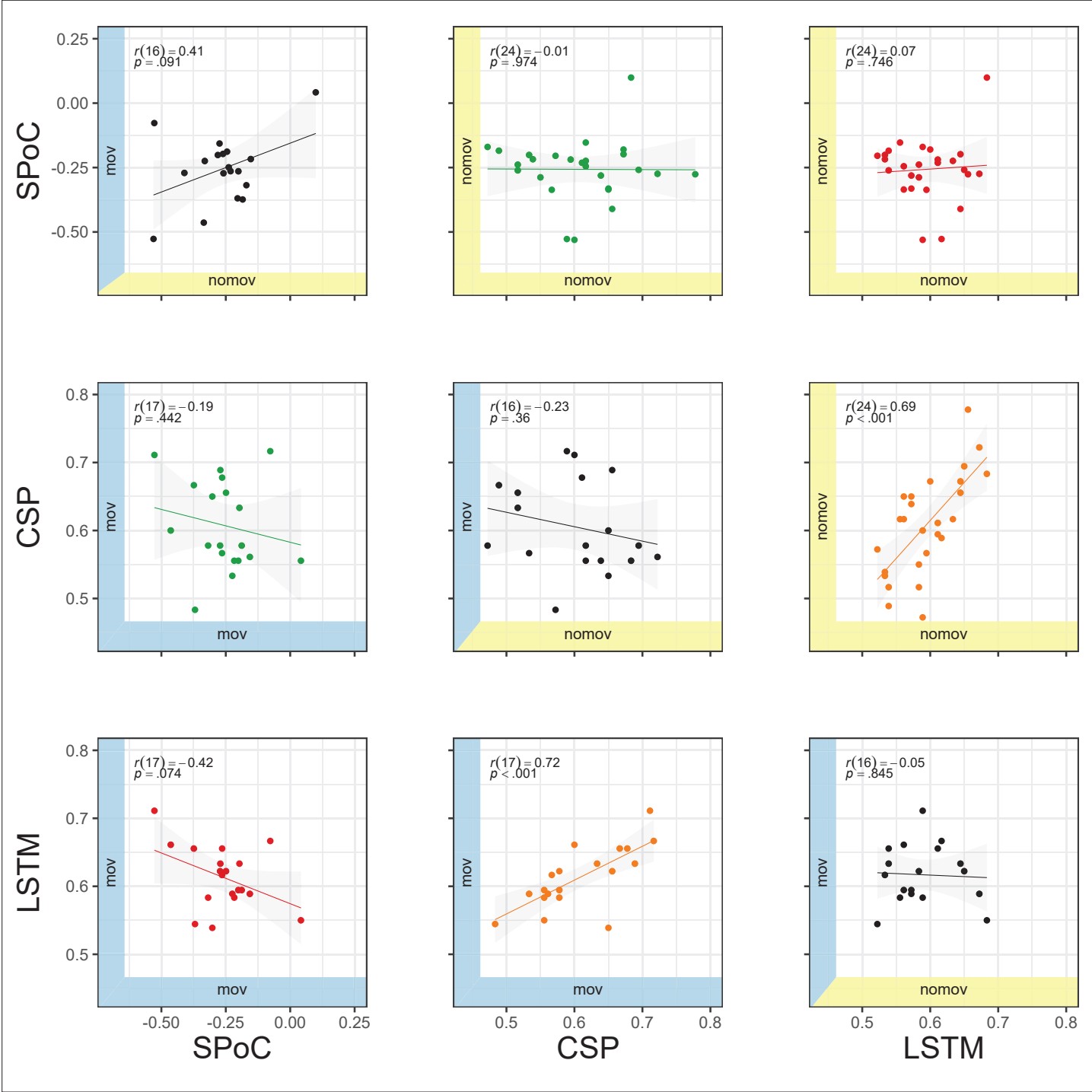

**Figure 10.** Correlation of performances across methods (source power comodulation [SPoC], common spatial pattern [CSP], long short-term memory [LSTM]) and conditions (nomov: without head movement, mov: with head movement). The model performance metrics are classification accuracy (CSP and LSTM) and correlation coefficients (SPoC; note: based on our hypothesis of an inverse relationship between emotional arousal and alpha power, more negative values indicate better predictive performance). Plots above and below the diagonal show data from the nomov (yellow axis shading, *upper right*) and the mov (blue axis shading, *lower left*) condition, respectively. Plots on the diagonal compare the two conditions (nomov, mov) for each method. In the top left corner of each panel, the result of a (Pearson) correlation test is shown. Lines depict a linear fit with the 95% confidence interval plotted in grey. The data underlying this figure can be downloaded as *Figure 10—source data 1*.

The online version of this article includes the following figure supplement(s) for figure 10:

**Source data 1.** Decoding results per decoding approach, movement condition, and participant.

methodological and conceptual limitations compared to SPoC and CSP, such as underestimating sources of noise, disregarding the generative model that underlies EEG data, and consequently a limited interpretability (for details, see *Dähne et al., 2014*). We therefore did not include this analysis in the final report.

## LSTM

Despite having recently gained more attention with the fast progress of deep learning (e.g., more efficient hardware and software implementations), LSTMs still need to stand up to well-established models such as CSP for EEG analysis. We found that the LSTM can extract features from neural input components that reflect changes in subjective emotional arousal and that the accuracy of its predictions in both conditions (nomov, mov) matched closely the ones of CSP (see *Figures 8 and 10*). It is noteworthy that for the CSP model, the (LDA-based) classification rested on narrowly defined spectral features of the signal while for the LSTM model, the input was the signal in the time domain and the feature selection process was part of the model fitting. The strong correlation between the predictions of the two models suggests that the LSTM extracts similar information as the CSP to make its prediction, namely power. Higher accuracies may be achievable with LSTM models due to their non-convex optimization landscape. However, in our two-step HP search, we found that for each subject a range of different HP settings led to similar prediction accuracies (see *Figure 4—source data 1*). Model ensembles, although computationally demanding, could further increase the robustness of the estimates (*Opitz and Maclin, 1999*; *Rokach, 2010*; *Dietterich, 2000*). Although it is often stated that deep learning models require large datasets (for an empirical perspective, see *Hestness et al., 2017*), our model, with its architecture of one to two LSTM layers followed by one to two fully connected layers, converged in less than 200 training iterations on a relatively small dataset. This quick convergence is partly due to the fast gradient flow through the memory cell of the LSTM during the weight update, which is an additional advantage of the LSTM over other RNNs (*Doetsch et al., 2014*; *Hochreiter and Schmidhuber, 1997*). Additionally, the spatial-spectral filtering in our study (i.e., SSD-based extraction of narrow-band alpha components) may have eased the training of the LSTM. With more data, an LSTM could be trained on raw data or longer segments of the EEG to preserve more of the continuous structure and ultimately exploit its central property, as a dynamic model, of detecting long-term dependencies in the input.

In contrast to SPoC and CSP, we did not compute explanatory topoplots or sources from the LSTM, since the analysis of predictions on input level in non-linear deep learning models constitutes a challenge in itself (i.e., 'black box' problem of deep learning). However, 'explainable artificial intelligence' (XAI) is an active area of research in machine learning, aiming to open this 'black box'. For EEG, there are attempts to create topologically informative maps in the signal space that explain the decision of simple shallow neural networks (*Sturm et al., 2016*). Also for the more complex LSTM model, XAI methods were applied, for example, on text data (*Arras et al., 2017*; see also *Lapuschkin et al., 2019*). However, exploring and validating these approaches on our data was beyond the scope of this study.

In summary, we find that SPoC, CSP, and LSTM can be used to decode subjective emotional arousal from EEG acquired during a naturalistic immersive VR experience. The source of the alpha oscillations could be localized in parieto-occipital regions.

Compared to other EEG decoding paradigms (e.g., lateralized motor imagery; *Herman et al., 2008*), the accuracy of our models was relatively low. This may be a consequence of (1) the fast-changing events in the VR experience (particularly the rollercoasters), (2) the asynchronicity of the two data streams as participants retrieved their emotional states from memory in retrospective ratings, (3) the generally high inter-individual variability in the interpretability of subjective self-reports (*Blascovich, 1990*), and (4) the 'single-trial' study design and its relatively short time series. With respect to (1)–(3), people's memory for feelings and events is susceptible to distortions and biases (*Kaplan et al., 2016*; *Levine and Safer, 2002*). Following *McCall et al., 2015*, we elicited the memory recall by showing participants an audiovisual replay of their experience from their own perspective in the VR headset while recording continuous ratings. This aimed to minimize biases related to the point of view (*Berntsen and Rubin, 2006*; *Marcotti and St Jacques, 2018*) or timescale (e.g., *Fredrickson and Kahneman, 1993*) during recall (as discussed in *McCall et al., 2015*). Lastly, while our research aimed to explore the role of the alpha frequency band in the

appraisal of emotional arousal (see Introduction), higher frequencies could carry additional information about the phenomenon leading to better model predictions. However, higher frequency bands also include non-neural (e.g., muscle activity-related) signals, limiting the interpretability of those results.

## Limitations

Our study has limitations that need to be considered when interpreting the results: while being engaging, emotionally arousing and tolerable for the subjects, the commercial content used for stimulation did not provide access to the source code in order to control and extract stimulus features (e.g., height or speed of the rollercoasters). In general, creating high-quality VR content is a challenge for research labs, but there are recent efforts to provide toolboxes that facilitate customized VR development and scientific experimentation in VR (e.g., *Grübel et al., 2017*; *Brookes et al., 2020*).

The length of the experience was chosen to minimize habituation to the stimulus and inconvenience caused by the recording setup (EEG electrodes and VR headset). This led to relatively short recording times per subject and condition. Data sparsity, however, is challenging for decoding models, which need a sufficient amount of data points for model training and evaluation, where especially larger training sets lead to more robust predictions (*Hestness et al., 2017*). We used cross-validation, which is commonly applied in scenarios of limited data, to achieve a trade-off between training and validation data (*Bishop, 2006*). Nevertheless, the models and results can be expected to perform more robustly with more training data.

We here confirm findings from static stimulation under more naturalistic conditions. To systematically investigate differences between approaches, a study with a within-subject design would be required. We hope that our study provides a stepping stone and motivation in this direction.

Finally, emotional arousal is a multi-faceted mind-brain-body phenomenon that involves the situated organism and its interaction with the environment. The training data for multivariate models such as the LSTM can include other modalities, such as peripheral physiological (e.g., HR, GSR) or environmental (e.g., optical flow) features. Naturalism can be further increased by sensorimotor interaction (beyond head movements) in immersive VR (*McCall et al., 2015*) or by mobile EEG studies in real-world environments (*Debener et al., 2012*), which, however, poses further challenges to EEG signal quality (*Gwin et al., 2010*).

## Conclusion

We conclude that different levels of subjectively experienced emotional arousal can be decoded from neural information in naturalistic research designs. We hope that combining immersive VR and neuroimaging not only augments neuroscientific experiments but also increases the generalizability and real-world relevance of neuroscientific findings.

## Acknowledgements

Thanks to Mina Jamshidi Idaji for her support on the EEG source reconstruction, to Nicolas Endres and Firat Sansal for valuable preparatory work, to Cade McCall for conceptual input on the study design, to Mert Akbal and Alireza Tarikhi for their help during data acquisition and data preprocessing, to Wojciech Samek, Klaus-Robert Müller, Frederik Harder, Kristof Schütt, Leila Arras, and Stefan Haufe for their methodological insights.

## Additional information

### Funding

| Funder | Grant reference number | Author |
|---|---|---|
| Bundesministerium für Bildung und Forschung | 13GW0206 | Felix Klotzsche Michael Gaebler |

| Funder | Grant reference number | Author |
|--------|------------------------|--------|
| Max Planck Society | Max Planck Society - Fraunhofer-Gesellschaft cooperation | Simon M Hofmann<br>Felix Klotzsche<br>Vadim Nikulin<br>Arno Villringer<br>Michael Gaebler |

The funders had no role in study design, data collection and interpretation, or the decision to submit the work for publication.

### Author contributions

Simon M Hofmann, Felix Klotzsche, Alberto Mariola, Conceptualization, Data curation, Formal analysis, Investigation, Methodology, Project administration, Software, Validation, Visualization, Writing – original draft, Writing – review and editing; Vadim Nikulin, Conceptualization, Formal analysis, Methodology, Supervision, Visualization, Writing – review and editing; Arno Villringer, Conceptualization, Funding acquisition, Resources, Supervision, Writing – review and editing; Michael Gaebler, Conceptualization, Formal analysis, Funding acquisition, Methodology, Project administration, Resources, Software, Supervision, Validation, Visualization, Writing – original draft, Writing – review and editing

### Author ORCIDs

Simon M Hofmann  http://orcid.org/0000-0003-0958-501X
Felix Klotzsche  http://orcid.org/0000-0003-3985-2481
Alberto Mariola  http://orcid.org/0000-0003-4660-1306
Michael Gaebler  http://orcid.org/0000-0002-4442-5778

### Ethics

Human subjects: Participants signed informed consent before their participation, and the study was approved by the Ethics Committee of the Department of Psychology at the Humboldt-Universität zu Berlin (vote no. 2017-22).

### Decision letter and Author response

Decision letter https://doi.org/10.7554/eLife.64812.sa1

## Additional files

### Supplementary files

• Transparent reporting form

### Data availability

We did not obtain participants' consent to release their individual data. Since our analyses focus on the single-subject level, we have only limited data which are sufficiently anonymized (e.g., summarized or averaged) to be publicly shared. Wherever possible, we provide "source data" to reproduce the manuscript's tables and figures (Figures 2, 4, 8 and 10). The scripts of all analyses are available at https://github.com/NeVRo-study/NeVRo (copy archived at https://archive.softwareheritage.org/swh:1:rev:669d5c2d6c73cbb70422efb933916fe8304195b5).

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

# Appendix 1

## Details of the rollercoasters

The 'Space' rollercoaster did not feature outstanding events during the ride besides two vertical spins starting around 47 and 73 s after the onset of the experience. Virtual collisions of asteroids floating through the scenery led to explosions of the celestial bodies involved, accompanied by an explosive sound. Apart from this, there were little sound effects during the space experience.

The 'Andes' rollercoaster included a steep drop (24 s after onset), two jumps with steep landings (31 and 67 s after onset), two passages through fires under the tracks (20 and 55 s after onset) and a looping (60 s after onset). Sound effects mimicked the sound of the waggon on the tracks, the fire, and the airflow. In the background a jingling melody was played.

## Simulator sickness questions

The wording and items to assess simulator sickness were:

Please rate on a scale from 1 to 7 how much each symptom below is affecting you right now:

1. General discomfort
2. Nausea
3. Dizziness
4. Headache
5. Blurred vision
6. Difficulty concentrating.

