## [Editor Report]

Hofmann et al., investigate the link between two phenomena, emotional arousal and cortical α activity. Although α activity is tightly linked to the first reports of electric activity in the brain nearly 100 years ago, a comprehensive characterization of this phenomenon is elusive. One of the reasons is that EEG, the major method to investigate electric activity in the human brain, is susceptible to motion artifacts and, thus, mostly used in laboratory settings. Here, the authors combine EEG with virtual reality (VR) to give experimental participants a roller coaster ride with high immersion. The ride, literally, leads to large ups and downs in emotional arousal, which is quantified by the subjects during a later rerun. Several different decoding methods were evaluated, and each showed above-chance levels of performance, substantiating a link between lower levels of parietal/occipital α and subjective arousal in a quasi-naturalistic setting.

---

## [Decision Letter]

**Decision letter after peer review:**

Thank you for submitting your article "Decoding subjective emotional arousal from EEG during an immersive Virtual Reality experience" for consideration by *eLife*. Your manuscript has been reviewed by 2 peer reviewers, and the evaluation has been overseen by Drs. Shackman (Reviewing Editor) and Baker (Senior Editor). The following individual involved in review of your submission has agreed to reveal their identity: Peter König (Reviewer #1).

Summary:

Hofmann et al., investigate the link between two phenomena, emotional arousal and cortical α activity. Although α activity is tightly linked to the first reports of electric activity in the brain nearly 100 years ago, a comprehensive characterization of this phenomenon is elusive. One of the reasons is that EEG, the major method to investigate electric activity in the human brain, is susceptible to motion artifacts and, thus, mostly used in laboratory settings. Here, the authors combine EEG with virtual reality (VR) to give experimental participants a roller-coaster ride with high immersion. The ride, literally, leads to large ups and downs in emotional arousal, which is quantified by the subjects during a later rerun. Three different decoding methods were evaluated (Source Power Comodulation, Common Spatial Patterns, and Long Short-Term Memory Recurrent Neural Networks), each of which demonstrated above-chance levels of performance, substantiating a link between lower levels of parietal/occipital α and subjective arousal in a quasi-naturalistic setting.

The reviewers both expressed some enthusiasm for the manuscript:

– The study is timely and makes an important contribution to our understanding of the relation of emotions and sensory processing.

– Of potentially great interest to a broad audience.

– The embedding in historic literature is excellent. I like it a lot.

– This work is notable because the roller-coaster simulation is a well-controlled, yet dynamic manipulation of arousal, and in its comparison of multiple decoding approaches (that can model the dynamics of affective responses). Indeed, this is an interesting proof of concept that shows it is possible to decode affective experience from brain activity measured during immersive virtual reality.

Major revisions:

Nevertheless, both reviewers expressed some significant concerns.

1. The result section is written as if there would be a preceding method section. But there is not. As a consequence, a large part of the results is incomprehensible without scrutinizing the method section further down. I'd suggest integrating about 2-3 pages worth of the total 17 pages method section into the result section proper. For example, you can safely assume that the majority of readers are not familiar with SPoC and CSP, might have heard of LSTM, but do not know the details relevant for the present context. Nonmov and mov conditions, well, might be guessed, but it is better to just state it. Similarly, Space coaster and Andes coaster, yeah, I just assumed that these are two different sections in the lab-amusement park. Still, make a short comment and reduce guessing.

2. Potential perceptual confounds: Decoding the emotional arousal vs. decoding break vs. ride. The emotional arousal drops drastically during the break. Ok, that's the nature of the break. However, the visual input also changes a lot during the break. This raises the possibility that the decoding emotional arousal largely rests on segmenting break vs. non-break periods by differentiating high motion visual input during the ride vs. static stimulation during the break. Figure 2 suggests some power differentiating different levels of emotional arousal within the ride intervals, but it is not really obvious. To address this issue, can you either better document and visualize the data or supply an analysis, not including the break, and just decoding variations of emotional arousal during the ride sections.

3. Temper the framing and claims to better align with the actual data and approach. The authors advocate that naturalistic experiments are needed to study emotional arousal, because "static" manipulations are not well-suited to capture the continuity and dynamics of arousal. This point is well-taken, but no comparisons were made between static and dynamic methods. Thus, although the work succeeds in showing it is possible to use machine learning to decode the subjective experience of arousal during virtual reality, it is not clear what new insights naturalistic manipulations and the machine learning approaches employed have to offer. Here are some suggestions for empirically evaluating the importance of dynamics in characterizing arousal:

a. Compare the effectiveness of the models developed in the present study against more conventional measures of arousal. Does a standard measure of occipital/parietal α predict subjective ratings as well as more complex models?

b. Experimentally compare correlates of arousal in static vs naturalistic manipulations. Do models trained to predict the arousal of participants during VR generalize to standard tasks and vice versa?

c. Investigate the importance of temporal dynamics in modeling arousal. LSTM models can model temporal dynamics through recurrent connections. Does disrupting this aspect of LSTM models reduce performance? Are dynamic models necessary to explain these naturalistic data?

4. The methods used to assess model performance are also a concern. Decoding models were evaluated separately for each subject using 10-fold cross-validation, and inference on performance was made using group-level statistics. Because time-series data are being decoded, if standard cross-validation was performed the results could be overly optimistic. Additionally, hyperparameters were selected to maximize model performance which can also lead to biased estimates. This is particularly problematic because overall decoding performance is not very high. Here are some suggestions for evaluating model performance:

a. Use rolling cross-validation or larger blocks to confirm that autocorrelation in EEG data and arousal ratings does not bias results.

b. Perform inference using block permutation (e.g., by iteratively scrambling consecutive blocks of subjective arousal ratings and refitting models).

c. Use nested cross-validation to optimize hyperparameters to provide a less biased estimate of performance.

---

## [Author Response]

Major revisions:1. The result section is written as if there would be a preceding method section. But there is not. As a consequence, a large part of the results is incomprehensible without scrutinizing the method section further down. I'd suggest integrating about 2-3 pages worth of the total 17 pages method section into the result section proper. For example, you can safely assume that the majority of readers are not familiar with SPoC and CSP, might have heard of LSTM, but do not know the details relevant for the present context. Nonmov and mov conditions, well, might be guessed, but it is better to just state it. Similarly, Space coaster and Andes coaster, yeah, I just assumed that these are two different sections in the lab-amusement park. Still, make a short comment and reduce guessing.

Given the methodological focus of the manuscript, we restructured the manuscript, moving the Methods and Materials section between the Introduction and the Results, as suggested by *eLife*. The models (SPoC, CSP, and LSTM) are now briefly introduced in the Introduction and detailed in the Methods section. We further made sure that the terms “Space” and “Andes” as well as “nomov” and “mov” condition are clearly introduced and explained.

2. Potential perceptual confounds: Decoding the emotional arousal vs. decoding break vs. ride. The emotional arousal drops drastically during the break. Ok, that's the nature of the break. However, the visual input also changes a lot during the break. This raises the possibility that the decoding emotional arousal largely rests on segmenting break vs. non-break periods by differentiating high motion visual input during the ride vs. static stimulation during the break. Figure 2 suggests some power differentiating different levels of emotional arousal within the ride intervals, but it is not really obvious. To address this issue, can you either better document and visualize the data or supply an analysis, not including the break, and just decoding variations of emotional arousal during the ride sections.

Following the reviewers’ suggestion, we ran the decoding and inference procedures (with the “sub-blocked” cross-validation, see Reviewer comment 4, and permutation procedures of the new supplementary analyses) once *with* and once *without* the data from the break. We did this for SPoC and CSP, skipping LSTM due to its computational processing cost and duration, and the comparable performance with CSP in the original analyses.

For both models (SPoC and CSP) and in both head movement conditions (nomov, mov), the decoding performance decreased for data *without* compared to data *with* the break (for CSP in the nomov condition significantly so; *t*(25) = 2.23, *p* = .034) but all previous findings remained significantly above chance. CSP results are visualized in Author response image 1. (For comparison, Author response image 2 shows the classification results of the linear model, logistic regression, which is introduced in Reviewer comment 3a).

**Author response image 1. sa2fig1:** CSP decoding performance shown as distributions of the means of the permuted null distributions (blue) over all subjects next to the distribution of the original, unpermuted decoding scores (yellow/green) on data *with* (left column) and *without* the break (right column) as well as with (lower row, “mov”) and without (upper row, “nomov”) free head movement. One-sided paired *t*-tests indicate that the means of these distributions (dotted vertical lines) differ significantly, with higher performance when decoding from the unpermuted arousal ratings for all conditions (ns: not significant, *: *p* <.05, **: *p* <.01, ***: *p* <.001).

This supplementary analysis is described and discussed in the revised manuscript as detailed below:

Methods and Materials / Data analysis / Supplementary analyses: methods:

“Excluding the break for model training

“The 30-s break differed from the rollercoaster rides in visual features (e.g., static vs dynamic input) and in arousal ratings, which were constantly relatively low during the break (see Figure 5). […]To test whether excluding the break changed the model performance, we compared the distributions of the decoding performance parameters (SPoC: Pearson correlation with target; CSP: ROC-AUC) from the data with and without the break using two-sided paired *t*-tests. We did this per model and movement condition.”

Results / Neurophysiology / Supplementary analyses: results:

“Excluding the break for model training

SPoC and CSP performed significantly above chance level also when trained and tested on data without the break section. […] Removing the break from the training data overall numerically decreased the decoding performances of both models. For CSP, the decrease was significant in the nomov (*t*(25) = 2.23, *p*=.034) and not significant in the mov condition (*t*(18) = 0.57, *p*=.58). For SPoC, the decrease (Pearson correlation) was not significant in both conditions (nomov: *t*(25) = -1.66, *p*=.108; mov: *t*(18) = -1.13, *p*=.269).”

Discussion:

“Moreover, to test the impact of the differences between the rollercoasters and the break, for example regarding visual dynamics and elicited emotional arousal, on the decoding performance, SPoC and CSP analyses were repeated on the data without the break. Again, the average decoding performances decreased compared to the data with the break, but remained significantly above chance level for both head movement conditions. The decrease in decoding performance with the break removed may result from (a) less training data being available and (b) a narrower range of emotional arousal values, more similar classes ("high arousal" and "low arousal"), and therefore a more difficult distinction.”

3. Temper the framing and claims to better align with the actual data and approach. The authors advocate that naturalistic experiments are needed to study emotional arousal, because "static" manipulations are not well-suited to capture the continuity and dynamics of arousal. This point is well-taken, but no comparisons were made between static and dynamic methods. Thus, although the work succeeds in showing it is possible to use machine learning to decode the subjective experience of arousal during virtual reality, it is not clear what new insights naturalistic manipulations and the machine learning approaches employed have to offer. Here are some suggestions for empirically evaluating the importance of dynamics in characterizing arousal:

Below, we detail the changes in the revised manuscript (tempering the framing) together with the additional analyses we ran. In general, we don’t want to claim that studying emotions in more naturalistic conditions works better than in static frameworks but that it *also* works under more naturalistic circumstances for which an increased level of face validity (i.e., generalizability towards the experience outside of the laboratory) can be assumed.

The main excerpts from the revised manuscript are:

Introduction: “This may be particularly true for affective phenomena like emotions.”

Discussion: “While such stimuli provide a high degree of experimental control in terms of low-level properties and presentation timings, the emotional experience and its neurophysiology under event-related stimulation may differ from the emotional experience in real-life settings, which is perceptually complex, multisensory, and continuously developing over time.”

a. Compare the effectiveness of the models developed in the present study against more conventional measures of arousal. Does a standard measure of occipital/parietal α predict subjective ratings as well as more complex models?

As a “standard measure”, we used a linear model (logistic regression) to predict the subjective level of (high vs low) emotional arousal from α power in an occipital/parietal region-of-interest (average of electrodes Pz, P3, P7, P4, P8, O1, O2, OZ): First, the EEG of each participant was filtered in the α band (±2 Hz around each participant’s α peak) via the EEGLAB (v.2019.1) ‘*pop_eegfiltnew.m’* function in MATLAB (v.R2019a). After a Hilbert transformation (*‘hilbert.m’*), α power was extracted by squaring the signal. The resulting time series was resampled to 1 Hz by averaging the power in non-overlapping 1-s epochs. To test the effect of autocorrelations in the data, training and testing of the logistic regression followed the same “sub-blocked” cross-validation regime as reported for CSP in response to Reviewer comment 4a (see below). The full logistic regression procedure was run in Python (v.3.8.5) via scikit-learn (v.0.24.1; Pedregosa et al., 2011).

On the group level, decoding performance (ROC-AUC) was significantly above chance level (1000 permutation on target labels) for both conditions when including break: nomov (*t_nomov_*(25) = -3.66, *p_nomov_* <.001) and mov (*t_mov_*(18) = -2.94, *p_mov_* <.01).

Conversely, when excluding the break, the decoding performance remained significant only for the mov condition (*t_mov_*(18) = -2.06, *p_mov_* <.05), but not for the nomov condition (*t_nomov_*(25) = -0.68, *p_nomov_* = .493; see Author response image 2).

**Author response image 2. sa2fig2:** Logistic regression decoding performance across subjects was significantly above chance level (permuted arousal ratings with 1000 permutations) in all but the condition without head movement and without the break. (For more details, please refer to the caption of Author response image 1).

The number of participants, for which states of high and low arousal could be correctly classified with logistic regression in both head movement conditions (nomov, mov; including break) are shown and compared to CSP in Author response table 1. The results further support the association between parieto-occipital α power and emotional arousal and show that logistic regression can detect this relationship.

**Author response table 1. sa2table1:** Number of participants, for which the level of emotional arousal (low, high) could be significantly predicted from α power (on data including break).

Significant prediction	nomov	mov
Logistic regression	5/26	4/19
CSP	9/26	5/19

However, logistic regression also has limitations, compared to approaches such as SPoC and CSP. Most importantly, linearly regressing power values (i) underestimates especially strong sources of noise, (ii) disregards the generative model that underlies EEG data and which describes the mapping from neural sources to recorded signals. This is because the nonlinear operator “squaring” is performed for each sensor before finding spatial weights, and (iii) makes the results difficult to interpret in neurophysiological terms due to channel-specific biases (27).

b. Experimentally compare correlates of arousal in static vs naturalistic manipulations. Do models trained to predict the arousal of participants during VR generalize to standard tasks and vice versa?

As mentioned above, we investigated whether findings from more static emotional stimulation (association of parieto-occipital α power and emotional arousal) extend to more naturalistic manipulations.

We agree that testing how and where there are differences between findings from naturalistic studies and from static experiments will be a very important field of study. To approach it while not changing multiple factors (e.g, task, participants, session), one would ideally acquire new data in a within-subject design that includes both a dynamic and a static emotion induction, which is – unfortunately – beyond the scope of our study.

We hope that our study serves as a stepping stone and motivation in this direction.

We added the following paragraph to the revised Limitations section:

“We here confirm findings from static stimulation under more naturalistic conditions. To systematically investigate differences between approaches, a study with a within-subject design would be required. We hope that our study provides a stepping stone and motivation in this direction.”

c. Investigate the importance of temporal dynamics in modeling arousal. LSTM models can model temporal dynamics through recurrent connections. Does disrupting this aspect of LSTM models reduce performance? Are dynamic models necessary to explain these naturalistic data?

Our study does not allow us to sufficiently answer whether dynamic models are necessary to explain these naturalistic data as this capacity of LSTMs is not fully taken advantage of due to the small size of our dataset. To gain enough training samples for the model from this small dataset, we split the sequences into short segments of 1-sec length, which limited the LSTM to apply its full potential and extract mid- to long-term dynamics in the EEG signal which are expected to unfold along the arousing VR experience. Moreover, switching off recurrent connections at the LSTM gating units would raise the question whether changes in decoding performance are caused by the disabled dynamic capacity, or, by modifications in the number of weights at the gating units. Despite these empirical and conceptual challenges, we agree that a systematic analysis of LSTMs and their dis/advantages for EEG analysis would be a valuable contribution. We hope that the comparison to the non-dynamic CSP and the additional analysis including the linear model (see Reviewer comment 3a) show that, in our study design, LSTMs and their capacity to model temporal dynamics were not superior.

We highlight the feature of LSTM as a dynamic model and the limitation to fully exploit this feature in our study design by adding the following fragment to the revised Discussion:

“With more data, an LSTM could be trained on raw data or longer segments of the EEG to preserve more of the continuous structure and ultimately exploit its central property, as a dynamic model, of detecting long-term dependencies in the input.”

4. The methods used to assess model performance are also a concern. Decoding models were evaluated separately for each subject using 10-fold cross-validation, and inference on performance was made using group-level statistics. Because time-series data are being decoded, if standard cross-validation was performed the results could be overly optimistic. Additionally, hyperparameters were selected to maximize model performance which can also lead to biased estimates. This is particularly problematic because overall decoding performance is not very high. Here are some suggestions for evaluating model performance:

The reviewers’ concrete suggestions led us to perform the control analyses detailed below. We report these in the Methods and in the Results section of the revised manuscript, together with the original results.

a. Use rolling cross-validation or larger blocks to confirm that autocorrelation in EEG data and arousal ratings does not bias results

We applied an adaption of rolling cross-validation (CV), “sub-blocked” CV, to the binary classification of CSP (LSTM-based analysis was skipped due to its computational processing cost and duration, and the comparable performance with CSP in the original analyses).

Please note that the originally reported performance metric accuracy (proportion of correctly classified samples) would be biased for unbalanced classes. In the supplementary analyses that we added to the manuscript, we therefore used the area under the curve of the receiver operating characteristic (ROC-AUC) instead.

The manuscript was revised accordingly in combination with reviewers’ comment 4b, which follows:

b. Perform inference using block permutation (e.g., by iteratively scrambling consecutive blocks of subjective arousal ratings and refitting models)

As suggested by the reviewers, we ran a supplementary analysis using block permutation to obtain assumption-free null distributions that reflect the autocorrelative nature of the data. The supplementary results for the CSP-based decoding (as reported in the revised manuscript, see above) overall support the results of the original analysis: Also when removing the break and when taking the autocorrelative structure in the data into account, our models were able to decode the levels of emotional arousal above chance level, reducing the probability that the original results are explained by artefacts or confounds.

The methods, results and discussion of the supplementary analyses (Reviewer comment 4a and b) are described in the revised manuscript as follows:

Methods and Materials / Data analysis / Supplementary analyses: methods:

**“**Sub-blocked cross-validation and block permutation

For non-stationary, auto-correlated time-series data, randomized cross-validation can inflate the decoding performance (Roberts et al., 2017). […] On the group level, one-sided paired *t*-tests were used to compare the distribution of the actual decoding results against the distribution of the means of the null distributions per subject.”

Results / Neurophysiology / Supplementary analyses: results:

“Sub-blocked cross-validation and block permutation

To test for potential biases from the model or the data, specifically its auto-correlative properties, we ran the same analysis for CSP as above using sub-blocked chronological cross-validation and block permutation for statistical evaluation on the single-subject level. […] On the single-subject level (as assessed by permutation tests), decoding performance was significantly (p <.05) higher when decoding the actual, unpermuted labels compared to the block-permuted labels for 9/26 (34.62 %) participants in the nomov and 5/19 (26.32 %) participants in the mov condition.”

Discussion:

“To test for confounds or analytic artefacts, for example due to autocorrelations in the data, we additionally applied “sub-blocked” cross-validation for model training and block permutation for statistical evaluation. Also under these more strict evaluation conditions, the average decoding performance was significantly above chance level. It is therefore unlikely that the results can be explained solely by dependencies in the data (e.g., autocorrelation) which are not explicitly modelled in the main analysis.“

c. Use nested cross-validation to optimize hyperparameters to provide a less biased estimate of performance

We agree that nested cross-validation (CV) would be the best approach to optimize hyperparameters (HPs). However, the sparse data per subject (180 samples in the classification task) was not sufficient to extract a reliable estimate of optimal HPs via nested CV, due to the negative bias (i.e., pessimistic model fit) that CV alone induces on small datasets (Section 5.1. In Arlot and Celisse, 2010; Varoquaux et al., 2017). As an alternative, we performed a random HP search on a subset of subjects (n=10), with the aim to make the found HPs more generalizable (if they lead to similar performances when applied on all subjects).

The results suggest that the model performance is relatively robust across the narrow set of HPs within subjects: Author response image 3 and 4 show the performance distributions over the narrow set of pre-selected hyperparameters Hn in the nomov and mov condition, respectively. Shown are participants, for whose data the average accuracy over HP sets in Hn was (green) or was not (blue) significant: 9 out of originally 16 subjects in the nomov and 7 out of 16 subjects in the mov condition remained significant (*p* <.05, two-sided binomial tests against chance level). Here, testing significance over the average HP set accuracy is explicitly overly conservative due to the above-mentioned negative bias of individual HP sets and potential non-convergence of the training procedure caused by random weight initialization.

**Author response image 3. sa2fig3:** Accuracy over the narrow set of hyperparameter (HP) settings in the condition without free head movement (nomov). The average accuracy of HPsets was (*green*) or was not (*blue*) significant. *Orange*: Binomial distribution over random trials. *Black thick dotted line*: Significance threshold. *Red dots* indicate the originally significant subjects and their best performing HP set. If the mean of the HP-set distribution (*longer, dotted vertical line*) passes the *p*-value threshold (*shorter, black-dotted vertical line*), the average accuracy is significant (distribution becomes *green*, otherwise *blue*).

**Author response image 4. sa2fig4:** Accuracy over the narrow set of hyperparameter (HP) settings in mov conditions. For details, please see the caption of Author response image 3.